# SiBBlInGS: Similarity-driven Building-Block Inference using Graphs across States

## Abstract

Data in many scientific domains are often collected under multiple distinct states (e.g., different clinical interventions), wherein latent processes (e.g., internal biological factors) can create complex variability between individual trials both within single states and between states. A promising approach for addressing this complexity is uncovering fundamental representational units within the data, i.e., functional Building Blocks (BBs), that can adjust their temporal activity and component structure across trials to capture the diverse spectrum of cross-trial variability. However, existing methods for understanding such multi-dimensional data often rely on Tensor Factorization (TF) under assumptions that may not align with the characteristics of real-world data, and struggle to accommodate trials of different durations, missing samples, and varied sampling rates. Here, we present a framework for Similarity-driven Building Block Inference using Graphs across States (SiBBlInGS). SiBBlInGS employs a robust graph-based dictionary learning approach for BB discovery that considers shared temporal activity, inter- and intra-state relationships, non-orthogonal components, and variations in session counts and duration across states, while remaining resilient to noise, random initializations, and missing samples. Additionally, it enables the identification of state-specific vs. state-invariant BBs and allows for cross-state controlled variations in BB structure and per-trial temporal variability. We demonstrate SiBBlInGS on synthetic and several real-world examples to highlight its ability to provide insights into the underlying mechanisms of complex phenomena across fields.

## 1 Introduction

The analysis of high-dimensional time-series is increasingly important across various scientific disciplines, ranging from neuroscience (Kala et al., 2009; Mudrik et al., 2022) to social sciences (Jerzak et al., 2023) to genetics (Bar-Joseph et al., 2012; Tanvir Ahmed et al., 2023). These data, however, present a daunting challenge in terms of comprehensibility as they are often highly heterogeneous. Specifically, data in many domains are gathered under multiple states (e.g., clinical interventions), while latent factors may introduce variability across trials within states (e.g., internal biological processes that lead to variations in patient responses to treatment).

Current analysis methods often struggle to capture the full variability in such multi-state data. Additionally, integrating data from repeated trials within an observed state into a coherent representation is often challenged by variable session duration, sampling rates, or missing samples (Goris et al., 2014; Charles et al., 2018; Duncker & Sahani, 2018), and the common practice of within-state trial averaging, for example, obscures important patterns within individual trials.

A promising approach to analyzing multi-state data is by identifying fundamental representational units, i.e., Building Blocks (BBs), that are similar across trials and states (e.g., neural ensembles in the brain, social networks, gene groups), while exhibiting temporal activity variations from trial to trial. Identifying such cross-state BBs can facilitate the identification of the underlying latent processes and provide valuable insights into core commonalities and differences among states. However, uncovering these BBs poses a challenge, as their individual activities are often unobservable. Furthermore, it is plausible that in addition to trial to trial variations in the temporal activity of these BBs–subtle cross-state variations in their composition are presented across states. For instance, a neural ensemble may not only display temporal activity differences across different normal brain

activity sessions and seizures but also show subtle structural changes during seizures compared to normal activity (van den Berg & Friedlander, 2008), e.g., neurons that are not typically part of the ensemble might become involved during a seizure.

Here, we present SiBBlInG, a graph-driven framework to unravel the complexities of high-dimensional multi-state time-series data, by unveiling its underlying sparse, similarity-driven Building Blocks (BBs) along with their temporal activity. Our main contributions encompass:

- A novel framework that addresses challenges in real-world scenarios where dynamic structures evolve across time, states, and contexts.
- The ability to account for both cross-trial variation in temporal activities and subtle differences in the cross-state BB composition, as well as to accommodate varying trial conditions, including different time durations, sampling rates, missing samples, and trial counts.
- Demonstrations of our method's robustness through synthetic data examples and evaluations on multiple real-world datasets.

## 2 BACKGROUND AND RELATED WORK

Conventional 2D methods for identifying BBs underlying time-series often rely on Singular Value Decomposition (SVD) (Kogbetliantz, 1955), Principal Components Analysis (PCA) (Hotelling, 1933), Independent Components Analysis (ICA) (Hyvarinen et al., 2001), or Non-negative Matrix Factorization (NMF) (Lee & Seung, 1999), which prioritize identifying components based on maximum variability or independence, or strictly enforcing assumptions (e.g.,non-negativity), that may not align with the characteristics of some real-world data. Newer methods, such as Dynamic Mode Decomposition (DMD) (Schmid, 2010), further model the temporal dynamics more explicitly as dynamical systems. However, the design of these methods for 2D analysis makes their application to multi-state and multi-session data challenging. Tensor decomposition (e.g., PARAFAC (Harshman, 1970; Williams et al., 2018; Mishne et al., 2016)) and higher-order matrix decomposition (e.g., HOSVD (De Lathauwer et al., 2000)) offer alternatives to traditional 2D methods by considering trials as additional dimensions, thus accounting for higher-order tensors. Some generalizations of tensor factorization, such as neural tensor factorization (e.g., Wu et al. (2018; 2019)), introduce temporal dependencies to better model sequential relationships in the data. However, these methods fail to account for variability in trial duration, address state variability, or result in sparse $\ell_1$-driven basis BBs. Gaussian Process (GP) tensor factorization (GP-TF) (e.g.,Tillinghast et al. (2020); Wang & Zhe (2022b); Ahn et al. (2021); Xu et al. (2011); Zhe et al. (2016)) offer a significant advancement for integrating temporal dimensions into the factorization of high-dimensional data by combining GPs with tensor decomposition. Despite their capability in probabilistically modeling temporal interactions, GP-TF approaches cannot handle trials of varying durations or distinguish between within to between state variability. Some methods (e.g., Wang & Zhe (2022b)) address non-stationarity, but generally focus on more continuous temporal notions of non-stationarity. While some addressing sparsity, current GP-TF methods struggle to isolate interpretable, sparsely-distributed components based on co-activation, deviating from our paper's objectives. Significant modification to the GP kernels would be necessary to distinguish trials within states from those across states and account for the discrete nature of states vs trials.

All these methods, however, are inherently data-driven and do not leverage state meta-information. Targeted dimensionality reduction (TDR) (Mante et al., 2013) and model-based TDR (mTDR) (Aoi & Pillow, 2018; Aoi et al., 2020) directly regress rank-1 (TDR) or low-rank (mTDR) components to explicitly target task-relevant variables. However, they are not inherently adaptable to accommodate trials of varying duration or to differentiate between within-state and between-state variability.

Closer to our approach, dictionary learning (DL) (Olshausen & Field, 2004; 1996; Aharon et al., 2006), relying on robust theoretical foundations (e.g., Sun et al. (2016); Sulam et al. (2022)), reconstructs data points using a few vectors from a feature dictionary under a sparsity-promoting regularization term (e.g., LASSO) and often provides more interpretable representations than other methods(Tošić & Frossard, 2011). While traditional DL often treats individual data points as independent, recent DL models based on re-weighted $\ell_1$ (Candes et al., 2008; Garrigues & Olshausen, 2010) present sparsity regularization terms that account for spatio-temporal similarities between data points (Garrigues & Olshausen, 2010; Charles & Rozell, 2013; Charles et al., 2016; Zhang & Rao, 2011; Qin et al., 2017; Mishne & Charles, 2019). Re-Weighted $\ell_1$ Graph Filtering (RWL1-

GF) (Charles et al., 2022) was recently developed for demixing fluorescing components in calcium imaging recordings by learning a data-driven graph that redefines pixel similarity. While GraFT proves the efficacy of graph-based correlations in extracting meaningful features from imaging recordings, it is constrained to single-trial data and confines its graph construction to a single path–the pixel space of the data–overlooking the possibility of meaningful structures in other dimensions.

In the context of TF, some methods can be thought of as handling either identical BBs across states with flexible temporal patterns or fixed identical temporal traces across states with flexible cross-state BBs. In the shared response model (SRM) (Chen et al., 2015), a multi-subject fMRI model is proposed where the same temporal activity is applied to all individuals (states) who may have different spatial responses. However, SRM relies on the assumption of orthogonality for component identification, which may not align with biological plausibility. Multi-dataset low-rank matrix factorization (Valavi & Ramadge, 2019) assumes identical structure across datasets but requires pre-alignment and pre-processing, which may not always be practical. Fuzzy clustering (Yang, 1993), and similarly its multiview extension (Wei et al., 2020), and the wavelet tensor fuzzy clustering scheme (WTFCS) (He et al., 2018) allow data points to exhibit varying degrees of membership in multiple clusters, addressing limitations of methods that restrict data points to a single BB. However, this approach does not address varying trial durations or sampling rates, does not consider structural variations of BBs across views, focuses solely on BB structures rather than their temporal activities, and does not distinguish between within-state and between-state variability.

Notably, the above approaches are all constrained in their capacity to capture the intricacies of multi-state multi-trial variability. Particularly, they are either restricted by an orthogonality assumption (De Lathauwer et al., 2000; Chen et al., 2015), limited interpretability (De Lathauwer et al., 2000; Harshman, 1970), or inability to handle trials of different duration (De Lathauwer et al., 2000; Harshman, 1970; Williams et al., 2018; Mante et al., 2013; Aoi & Pillow, 2018; Aoi et al., 2020; Wei et al., 2020; He et al., 2018). Other methods require detailed labels and sufficient training data (Liu et al., 2022; Schneider et al., 2023), and cannot incorporate both within and between state variability (De Lathauwer et al., 2000; Harshman, 1970; Mante et al., 2013; Schmid, 2010; He et al., 2018), or to model cross-state variations in BB structure (De Lathauwer et al., 2000; Harshman, 1970; Williams et al., 2018; Valavi & Ramadge, 2019; Schmid, 2010). Hence, there is a need for new approaches that can provide a more comprehensive framework for identifying and exploring the BB's underlying high-dimensional multi-state data.

## 3 PROBLEM DEFINITION AND NOTATION

Consider a system with $N$ channels that collectively organize into a maximum of $p$ functional BBs, which represent groups of channels with shared functionality. These BBs serve as the fundamental constituents of a complex process, and their composition is not directly observed nor explicitly known. In particular, let $\boldsymbol{A} \in \mathbb{R}^{N \times p}$ capture these BBs in its columns, such that the value of $\boldsymbol{A}_{ij}$ describes the contribution of the $i$-th channel to the $j$-th BB, with a value of $0$ indicating that the channel does not belong to that BB. We assume that channels have the flexibility to belong to more than one BB, and that each BB is sparse (i.e., $\|\boldsymbol{A}_{:j}\|_0 = K << N \quad \forall j = 1 \dots p$).

We first consider a single instance of the system $\boldsymbol{Y} \in \mathbb{R}^{N \times T}$ (here termed "trial"), where $T$ time points is the duration of activity. During this trial, each BB exhibits temporal activity that influences the system's behavior, captured by $\boldsymbol{\Phi} \in \mathbb{R}^{T \times p}$, with the entry at index $(t, j)$ representing the activity of the $j$-th BB at time $t$. These temporal profiles are assumed to be smooth over time, bounded (i.e., $\|\boldsymbol{\Phi}\|_F < \epsilon_1$, with $\epsilon_1$ being a scalar threshold), and have low correlation between distinct BBs' activity (i.e., $\rho(\boldsymbol{\Phi}_{:j}, \boldsymbol{\Phi}_{:i}) < \epsilon_2 \quad \forall i \neq j$, with $\epsilon_2$ being a scalar threshold). In this single-trial scenario, our observations are limited to the combined activity of all BBs operating together, as captured by $\boldsymbol{Y} = \boldsymbol{A}\boldsymbol{\Phi}^T + \eta$, where $\eta$ is *i.i.d.* Gaussian observation noise.

In the more general setting, we observe a set of $M$ trials, $\{\boldsymbol{Y}_m\}_{m=1}^M$, where the duration of each trial may vary, i.e., $\boldsymbol{Y}_m \in \mathbb{R}^{N \times T_m}$, and the BBs ($\boldsymbol{A}$) remain constant across trials while their corresponding temporal activity ($\boldsymbol{\Phi}_m \in \mathbb{R}^{T_m \times p}$) may vary across trials to capture trial-to-trial variability. This general structure can typically be addressed by either concatenating the trials end-to-end and applying a matrix decomposition or by reshaping the trials to a uniform length (e.g., by time-wrapping (Venkatesh & Jayaraman, 2010) or zero-padding (Wang & Zhang, 2012)) and applying a tensor decomposition. However, both of these approaches overlook the significance of trial to trial

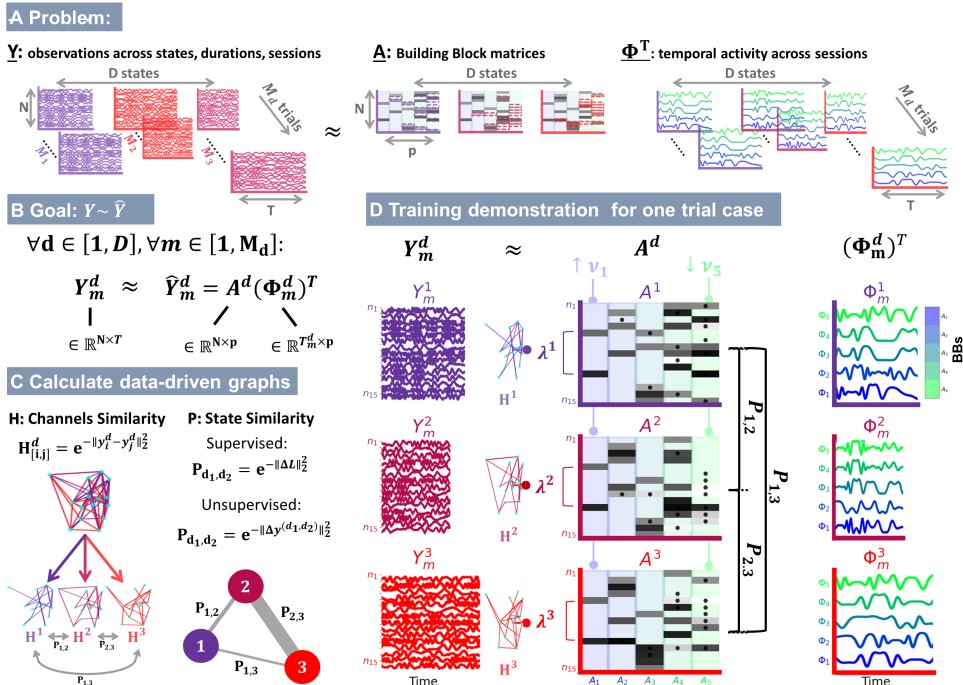

Figure 1: **The SiBBlInGS model. A** SiBBlInGS adapts to real-world datasets with varying session durations, sampling rates, and state-specific data by learning interpretable graph-driven hidden patterns and their temporal activity. **B** SiBBlInGS is based on a per-state-and-trial matrix factorization where the BBs ($A^d$) are identical across trials and similar across states. **C** BB similarity is controlled via data-driven channel graphs ($\boldsymbol{H}^d \in \mathbb{R}^{N \times N}$) and state-similarity graph ($\boldsymbol{P} \in \mathbb{R}^{D \times D}$), which can be either predefined (supervised) or data-driven. **D** The learning schematic with an exemplary trial for each of the 3 exemplary states. The BBs of each state $d$ (columns of $\mathbf{A}^d$) are constrained with two regularization terms: 1) state-specific $\boldsymbol{\lambda}^d$ captures similar activity between channels by leveraging a channel-similarity graph $\boldsymbol{H}^d$, and 2) $\boldsymbol{P}$, captures BB consistency across states via state similarity graph. $\boldsymbol{\nu}$ controls the relative level of cross-state similarity between BBs, allowing the discovery of both background and state-specific BBs. Higher (lower) $\boldsymbol{\nu}$ values promote greater (lesser) consistency of specific BBs across states (e.g. $\boldsymbol{\nu}_1$ v.s $\boldsymbol{\nu}_5$).

variability within and across states. Hence, the setting we focus on extends beyond a single set of trials; instead, we deal with a collection of $D$ multi-trial sets, each associated with a known state or condition. Within each $d$-th set, the number of trials, denoted as $M_d$, may also vary. In essence, the full observation set includes this collection of $D$ multi-trial sets, $\{\boldsymbol{Y}_m^1\}_{m=1}^{M_1}, ... \{\boldsymbol{Y}_m^D\}_{m=1}^{M_D}$, where each set represents a different state $d = 1 \dots D$, such that $\boldsymbol{Y}_m^d = \boldsymbol{A}^d(\boldsymbol{\Phi}_m^d)^T + \eta_m^d$.

We further assume that while the temporal activities of the BBs ($\boldsymbol{\Phi}_m^d$) can vary across trials, both within and between states, the compositions of the BBs themselves ($\boldsymbol{A}^d$) might also present subtle variations across different states. Specifically, we posit that the dissimilarity between BB compositions for any pair of distinct states $d$ and $d'$ ($d \neq d'$) reflects the dissimilarity between those states, such that the dissimilarity between $\boldsymbol{A}^d$ and $\boldsymbol{A}^{d'}$ for any distinct states $d$ and $d'$ is constrained by $\|\boldsymbol{A}^d - \boldsymbol{A}^{d'}\|_F < \epsilon_3(d, d')$. Here, $\epsilon_3(d, d')$ is a scalar threshold determined by the expected dissimilarity between these states (e.g., if considering different disease stages as states, we assume that consecutive disease stages are more similar to each other than they are similar to a healthy state, i.e., $\epsilon_3(d_{\text{disease}_1}, d_{\text{disease}_2}) < \epsilon_3(d_{\text{healthy}}, d_{\text{disease}})$).

The main challenge SiBBlInGS addresses is recovering the unknown underlying BBs ($\boldsymbol{A}^d$) and their associated temporal activity ($\boldsymbol{\Phi}_m^d$) for all states and trials given solely the combined activity observations (Fig. 1 top). Further notation details can be found in Section A.

# 4 THE SiBBLInGS MODEL

We present a framework to identify interpretable BBs based on shared temporal activity within trials and shared activation patterns across trials and states. Our framework serves as a foundation for understanding cross-trial and cross-state variability and enables deeper insights into how BBs differ across sessions both in terms of their structure and temporal dynamics. Unlike existing methods (Table 2), SiBBlInGS clusters BB components based on temporal similarity and enables the study of variability in high-dimensional data without assuming orthogonality. In addition, SiBBlInGS enables BB interdependency or overlap, recognizes the existence of variations in trial counts within states, and is capable of coping with trials of different duration or sampling rates. SiBBlInGS also offers both supervised and unsupervised state-similarity setting–thus providing the flexibility to choose between data-driven or predefined approaches based on the specific data structure.

We develop a dictionary learning-like iterative procedure that alternates between updating the BBs $\{\boldsymbol{A}^d\}$ and their temporal profiles $\{\boldsymbol{\Phi}_m^d\}$ for all states $d = 1 \ldots D$. Critical to our approach is the integration of both channels' nonlinear similarity knowledge and the understanding of how states differ. We thus augment the model with two graphs, one over channels, and one over states, to capture these relationships. The graph over channels is used to identify regularities between channels, and the states graph is used to promote consistency in BB structure across states. Mathematically, we formulate the fitting procedure as minimizing the following cost function $\{\widehat{\boldsymbol{A}}^d\}, \{\widehat{\boldsymbol{\Phi}}_m^d\}$ for all $d = 1 \ldots D$ and $m = 1 \ldots M_d$:

$$\min_{\{\boldsymbol{A}^d\}, \{\boldsymbol{\Phi}_m^d\}} \Sigma_d^D \left( \Sigma_m^{M_d} \left[ \|\boldsymbol{Y}_m^d - \boldsymbol{A}^d(\boldsymbol{\Phi}_m^d)^T\|_F^2 + \mathcal{R}(\boldsymbol{\Phi}_m^d) \right] + \mathcal{R}(\boldsymbol{A}^d) + \Sigma_{d' \neq d}^D P_{d,d'} \|(\boldsymbol{A}^d - \boldsymbol{A}^{d'})\boldsymbol{V}\|_F^2 \right)$$

where the first term is a data fidelity term, the second regularizes the BBs' temporal traces, the regularization $\mathcal{R}(\boldsymbol{A}^d)$ regularizes each BB to group channels based on shared temporal activity and to be sparse (as described in the next sections), and the last term regularizes BBs to be similar across states. The square matrix $\boldsymbol{P} \in \mathbb{R}^{D \times D}$ is a state-similarity graph that determines the effect of the similarity between each pair of states on the regularization of the distance between their BB representations. $\boldsymbol{P}$ can be set manually (supervised $\boldsymbol{P}$) or in a data-driven way (unsupervised $\boldsymbol{P}$), thus allowing selection based on specific goals, data type, and knowledge of data labels. Each of these two options offers unique benefits: the supervised variant enables explicit regulation of the similarity and the incorporation of important knowledge into the model based on human-expert familiarity with the data, whereas the unsupervised variant leverages the data itself to learn similarities and patterns, and is advantageous for learning data patterns without preconceived biases. The use of the matrix $\boldsymbol{V} = \text{diag}(\boldsymbol{\nu}) \in \mathbb{R}^{p \times p}$, accompanied by the weight vector $\boldsymbol{\nu} \in \mathbb{R}^p$, allows for assigning varying weights to individual BBs, thereby facilitating the creation of state-invariant vs state-specific BBs. The model operates iteratively, with updates applied to $\boldsymbol{A}^d$ and $\boldsymbol{\Phi}_m^d$ for each trial and state. The process is detailed in Algorithm 1 and in Figure 1, with computational complexity in Section E.

**Updating $\boldsymbol{A}^d$ :** In SiBBlInGS, we assume that BBs may require subtle state-to-state adaptations but are required to remain constant within a state. Hence, we demand that the BB matrix ($\boldsymbol{A}^d$) is shared across same-state trials but undergoes subtle adjustments across states, proportionate to the corresponding states' similarities. The update of $\boldsymbol{A}^d$ for each state $d$, is achieved via an extended re-weighted $\ell_1$ graph filtering with an integration of a channel-similarity graph in a way that promotes channels with similar temporal activity to be grouped into the same BBs. In particular, we update the $n$-th row of each $\widehat{\boldsymbol{a}}_n^d = \widehat{\boldsymbol{A}}_{n:}^d$ via the re-weighted procedure that alternates between updating $\widehat{\boldsymbol{a}}_{nj}^d$ and $\boldsymbol{\lambda}_{n,j}^d$ as:

$$\widehat{\boldsymbol{a}}_n^d = \arg\min_{\boldsymbol{a}_n^d} \|\boldsymbol{Y}_n^{d*} - \boldsymbol{a}_n^d(\boldsymbol{\Phi}^{d*})^T\|_2^2 + \Sigma_{j=1}^p \boldsymbol{\lambda}_{n,j}^d |\boldsymbol{a}_{n,j}^d| + \Sigma_{d' \neq d} \boldsymbol{P}_{dd'} \|(\boldsymbol{a}_n^d - \boldsymbol{a}_n^{d'}) \circ \boldsymbol{\nu}\|_2^2, (1)$$

with $\boldsymbol{\lambda}_{n,j}^d = \epsilon / (\beta + |\widehat{\boldsymbol{A}}_{n,j}^d| + w_{\text{graph}} |\boldsymbol{H}_{n:}^d \cdot \widehat{\boldsymbol{A}}_{:j}^d|)$, $\boldsymbol{Y}^{d*}$ is a matrix of size $N \times (\Sigma_{m=1}^{M_d} T_m^d)$ of horizontally concatenated observations from all $M_d$ trials of state $d$, and $\boldsymbol{\Phi}^{d*} \in \mathbb{R}^{(\Sigma_{m=1}^{M_d} T_m^d) \times p}$ is a matrix formed by vertically concatenating the current estimates of temporal traces from all trials of that state. Above, $\circ$ is element-wise multiplication, $\beta$, $\epsilon$, and $w_{\text{graph}}$ are model hyper-parameters. $\boldsymbol{H}^d$ and $\boldsymbol{P}^d$ are channel and state similarity graphs, described below:
**State similarity graph ($\boldsymbol{P} \in \mathbb{R}^{D \times D}$):** Can be either pre-defined (supervised) or data-driven. Here, we present the supervised version of $\boldsymbol{P}$, which is particularly useful when one has prior knowledge or

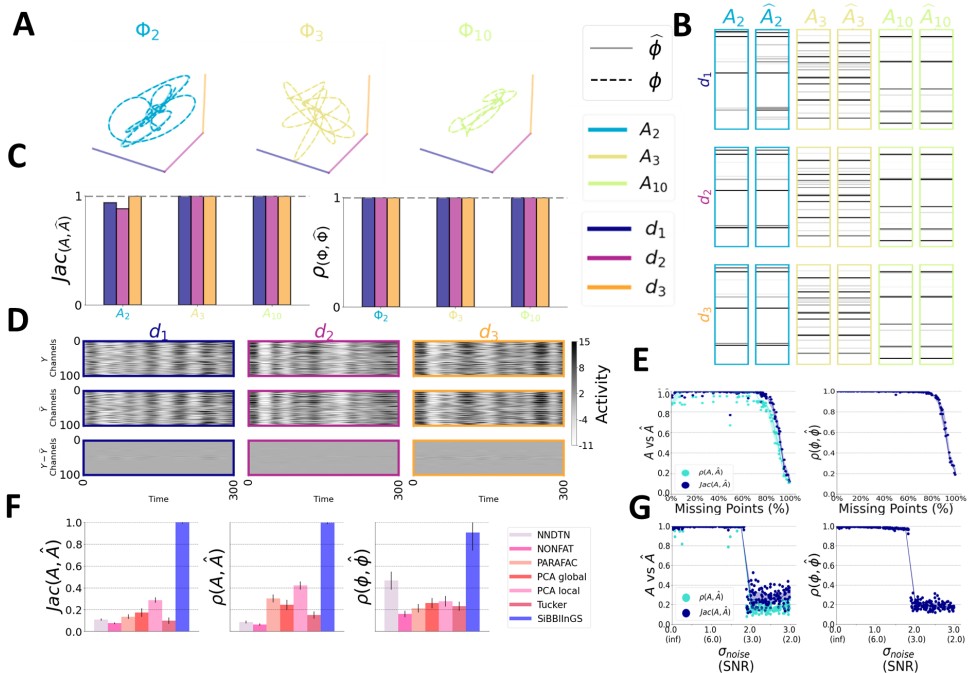

Figure 2: **Demonstration on Synthetic Data. A** Three example time traces identified by SiBBlInGS vs. ground truth traces, projected onto the axes of the three synthetic states. SiBBlInGS manages to recover both traces that are highly correlated with specific states (state-sensitive, e.g., $\mathbf{\Phi}_{10}$; green), as well as traces exhibiting similar activation across states (state-invaariant, e.g., $\mathbf{\Phi}_2$; blue). **B** Comparison between the identified BBs and the ground truth BBs. **C** Correlation between the identified time traces and the ground truth (right), and Jaccard index of the identified BBs compared to the ground truth (left). **D** Comparison between the ground truth data (top), SiBBlInGS reconstruction (middle), and the residual (bottom). **E** Performance under increasing levels of missing samples (200 repeats). The scattered dots represent model repetitions, the curves depict the median values calculated by rounding to the nearest 5%, and the background shading corresponds to the 25-75 percentiles. **F** Comparison to other relevant methods, including Tucker, PARAFAC, PCA "global" (applying a single PCA to all states), PCA "local" (applying PCA to each state), NONFAT Wang & Zhe (2022b), and NNDTN (discretetime NN decomposition with nonlinear dynamics, as implemented by Wang & Zhe (2022a)). See Section H.4 for details. **G** Performance under increasing levels of noise and random matrix initalizations (300 repeats). The scattered dots represent model repetitions, the curves depict the median values calculated by rounding to the nearest 0.25 noise std, and the background shading corresponds to the 25-75 percentiles. While the model remains robust under varying noise (SNR < 3), it experiences a phase transition at a specific noise level, aligning with dictionary-learning literature (e.g. Studer & Baraniuk (2012)).

expectations regarding quantitative state values that can be leveraged to integrate desired information into the model, while the data-driven approach is presented in Section B.2. This supervised version, unlike the data-driven option, assumes that a numerical label $\boldsymbol{L}_d$, associated with each state $d$, can provide valuable information for constructing the state-similarity graph $\boldsymbol{P}$ (e.g., vector labels that denote x-y positions in a reaching-out task where the states are the possible positions). This way, the similarity $\boldsymbol{P}_{d,d'}$ between each pair of states $(d, d')$ is calculated based on the distance between the labels ($\boldsymbol{L}_d, \boldsymbol{L}_{d'}$) associated with these states: $\boldsymbol{P}_{d,d'} = \exp\left(-\|\boldsymbol{L}_d - \boldsymbol{L}_{d'}\|_2^2/\sigma_{\boldsymbol{P}}^2\right)$ where $\sigma_{\boldsymbol{P}}^2$ controls how the similarities in labels scale to similarities in BBs. The supervised approach easily extends to both data with identical or different session duration, and can also handle categorical states as described in Sec. B.1.1.

**Channel similarity graph** ($\boldsymbol{H}^d \in \mathbb{R}^{N \times N}$): Defined by $\widetilde{\boldsymbol{H}}_{i,j}^d = \exp\left(-\|\boldsymbol{Y}_{i:}^{d*} - \boldsymbol{Y}_{j:}^{d*}\|^2/\sigma_{\widetilde{\boldsymbol{H}}}^2\right)$, where $\sigma_{\widetilde{\boldsymbol{H}}}$ is a model hyperparameter that controls the kernel bandwidth. To enhance the robustness of $\boldsymbol{H}^d$ for each state $d$, we utilize the previously computed state-graph ($\boldsymbol{P}$) to re-weight $\boldsymbol{H}$ along

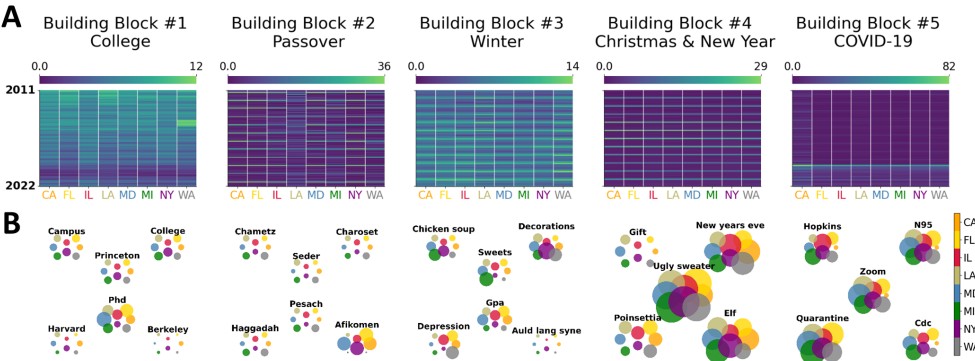

Figure 3: **Demonstration on Google Trends data. A** Temporal traces of the identified BBs demonstrate seasonal trends consistent with the words associated with each BB. **B** BBs by different states along with their per-state dominancy. States are marked by colors, and dot sizes represent the contribution of the term in the associated BB.

the states dimension. This process mitigates the influence of outliers and encourages the clustering of similarly behaving channels into the same BB. We achieve this by calculating $\boldsymbol{H}^d$ as a weighted sum $\Sigma_{d=1}^D \boldsymbol{P}_{d,d'} \widehat{\boldsymbol{H}}^{d'}$, and then retaining only the $k$ largest values in each row, setting the remainder to zero. We then symmetrize this graph and row-normalize it so that its rows sum to 1 (Sec. C). The advantage of this graph-driven re-weighted approach, compared to other TF and dictionary learning procedures, lies in that the weighted evolving regularization term ($\boldsymbol{\lambda}^d \in \mathbb{R}^{N \times p}$) is a function of the channel similarity graph $\boldsymbol{H}^d$ in a way that promotes the grouping (separating) of channels with similar (dissimilar) activity into the same (different) BBs. Specifically, as seen in the last term of the $\boldsymbol{\lambda}_{n,j}^d$'s denominator, for a given state $d$, a strong (weak) correlation between the similarity values of the $n^{th}$ channel ($\boldsymbol{H}_{n:}^d$) and the $j$-th BB ($\widehat{\boldsymbol{A}}_{:j}^d$) results in decreased (increased) $\boldsymbol{\lambda}_{n,j}^d$. Consequently, the $\ell_1$ regularization on $\widehat{\boldsymbol{a}}_n^d$ is reduced (increased), promoting the inclusion (exclusion) of this channel in the $j$-th BB.

After each update of all rows in $\boldsymbol{A}^d$, each column is normalized to have a maximum absolute value of 1. In practice, we update $\boldsymbol{A}$ ( equation 1) for a random subset of trials in each iteration to improve robustness and computation speed.

**Updating $\boldsymbol{\Phi}_m^d$:** The update step over $\boldsymbol{\Phi}_m^d$ uses the current estimate of $\boldsymbol{A}^d$ to re-estimate the temporal profile matrix $\boldsymbol{\Phi}_m^d$ independently over each state $d$ and trial $m$. Note that we do not enforce similarity in $\boldsymbol{\Phi}_m^d$ to allow for flexibility in capturing differences across states and trials. Thus, for each trial $m$ and state $d$, $\boldsymbol{\phi} = \boldsymbol{\Phi}_m^d$ is updated by solving the following minimization problem:

$$\widehat{\boldsymbol{\phi}} = \arg\min_{\boldsymbol{\phi} \geq 0} \left\| \boldsymbol{Y}_m^d - \boldsymbol{A}^d \boldsymbol{\phi}^T \right\|_F^2 + \gamma_1 \|\boldsymbol{\phi}\|_F^2 + \gamma_2 \|\boldsymbol{\phi} - \widehat{\boldsymbol{\phi}}^{\text{iter}-1}\|_F^2 + \gamma_3 \|\boldsymbol{\phi} - \boldsymbol{\phi}^{t-1}\|_F^2 + \gamma_4 \mathcal{R}_{\text{corr}}(\boldsymbol{\phi}) \quad (2)$$

where the first term is for data fidelity, the second term regularizes excessive activity, the third term encourages continuity across iterations ($\widehat{\boldsymbol{\phi}}^{\text{iter}-1}$ refers to $\boldsymbol{\phi}$ from the previous model iteration), and the fourth term is a diffusion term that promotes temporal consistency of the dictionary across consecutive samples ($\boldsymbol{\phi}^{t-1}$ refers to a shifted version of $\boldsymbol{\phi}$ by one time point), and $\mathcal{R}_{\text{corr}}(\boldsymbol{\phi}) = \left\| \left( \boldsymbol{\phi}^T \boldsymbol{\phi} - \text{diag}(\boldsymbol{\phi}^T \boldsymbol{\phi}) \right) \circ \boldsymbol{D} \right\|_{sav}$ promotes decorrelation among the temporal traces of BBs (where $\boldsymbol{D} \in \mathbb{R}^{p \times p}$ is a normalization matrix with $\boldsymbol{D}_{ij} = \frac{1}{\|\boldsymbol{\phi}_{:i}\|_2 \|\boldsymbol{\phi}_{:j}\|_2}$, and $sav$ stands for sum-of-absolute-values). This update step thus seeks to improve the dictionary by minimizing the cost function, while balancing sparsity, decorrelated elements, continuity, and temporal consistency (refer to Section D for how to solve equation 2 in practice).

## 5 EXPERIMENTS

**SiBBlInGS recovers ground truth BBs in synthetic data:** Synthetic data were generated with $D = 3$ states, each consisting of a single trial, with $p = 10$ ground-truth BBs, and $N = 100$ channels. Each $i$-th BB was generated with a maximum cardinality of $\max_{d,i} \|A_{:,i}^d\|_0 = 21$ channels, and

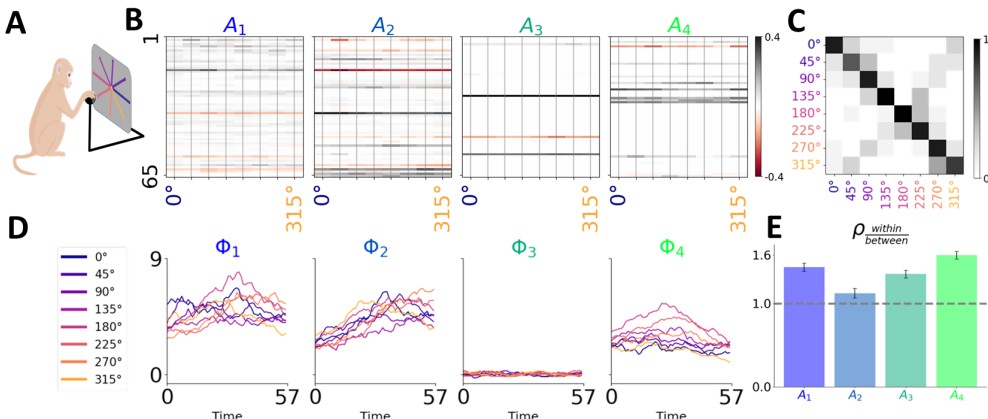

Figure 4: **Identification of Temporal Patterns in Monkey Somatosensory Cortex. A** The reaching out task (Rodriguez (2023)). **B** Sparse clusters of neurons representing identified BBs. **C** Confusion matrix of a logistic regression model using the inferred temporal traces to predict the state label. **D** The BB's temporal traces as they vary across states and time. **E** Ratios of within-to-between states temporal correlations for each BB, with $\frac{\rho_{within}}{\rho_{between}} > 1$, indicating states distinguishability.

on average, each channel was associated with 2.1 BBs. While the BBs were designed to be non-orthogonal, we constrained their pairwise correlations to be below a threshold of $\max \rho < 0.6$. The temporal dynamics of the synthetic data were generated by summing 15 trigonometric functions with different frequencies (Sec. H.2 for details). SiBBlInGS exhibited a monotonically increasing performance during training (Fig. 6A,B,C,D), and at convergence was able to successfully recover the underlying BBs in the synthetic data and their temporal traces (Fig 2A, B, C). Example traces demonstrate a high precision of the recovered temporal traces, with correlation to the ground truth traces being close to one (Fig. 2A, C, 6F). Furthermore, the identified BB components align closely with the ground truth BBs (Fig. 2B,C), as indicated by high Jaccard index values. Notably, tensor decomposition models were unable to identify the BBs nor their traces (Fig. 2F, Fig 6F).

**SiBBlInGS finds interpretable BBs in Google Trends data:** We use Google Trends Google Trends (Accessed 11 November 2022) to demonstrate SiBBlInGS' capability in identifying temporal and structural patterns by querying search term frequency on Google over time. We used a monthly Trends volume of 44 queries (fromJan. 2011 to Oct. 2022) related to various topics, as searched in 8 US states selected for their diverse characteristics Coulby (2000). The $p = 5$ BBs identified by SiB-BlInGS reveal meaningful clusters of terms, whose time traces convey the temporal evolution of user interests per region (Fig. 3A), while aligning with the seasonality of the BBs'components (Fig. 3B). For instance, the first BB represents college-related terms and shows a gradual annual decrease with periodic activity and a notable deviation during the COVID pandemic, possibly reflecting factors such as the shift to remote learning (Fig. 9, Sec. I.3). The 2-nd and 4-th BBs, respectively, demonstrate periodic patterns associated with Passover in April (Fig. 10, Sec. I.4) and winter terms in December. Interestingly, CA, FL, and NY—all states with larger Jewish populations—show more pronounced peaks of the "Passover" BB activity in April (when Passover is celebrated) compared to the other states (Fig. 10). The last BB represents COVID-related terms and exhibits temporal patterns with a sharp increase around Jan. 2020, coinciding with the onset of the COVID pandemic in the US. Remarkably, 'Hopkins' exhibits a less pronounced COVID-related search peak in MD (blue), where the university and hospital are located, likely attributed to its well-established local presence. Conversely, other states witnessed a more significant surge in Hopkins-related searches at the onset of the COVID outbreak, as Hopkins suddenly garnered increased attention during this period. This emphasizes our model's interpretability and the need to capture similar yet distinct BBs across states, distinguishing it from other tensor factorization methods (Fig. 8).

**SiBBlInGS identifies meaningful patterns in brain recordings:** We next test SiBBlInGS on neural activity recorded from the somatosensory cortex in a monkey performing a reaching task, as described by Chowdhury & Miller (2022). The data consist of 8 different hand angle directions, representing distinct states, with each angle comprising 18 trials observed under noisy conditions.

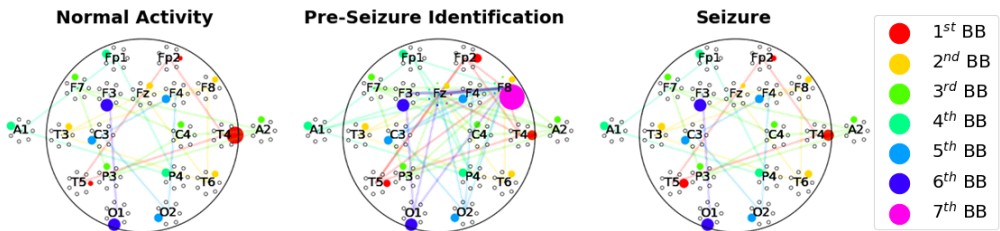

Figure 5: **Emerging local BBs in Epilepsy**. The recovered BBs under 1) normal activity, 2) activity during the 8 seconds proceedings CPS seizures located around the F8 area, and 3) activity during the seizures. Colors represent different BBs, and the size of the dots corresponds to the contribution of the respective electrode to each BB. Refer to Figure 13 for a comparison to other methods.

This spikes data were convolved over time with a Gaussian kernel. When applying SiBBlInGS with a maximum of $p = 4$ BBs, SiBBlInGS identified sparse functional BBs (Fig. 4B) along with meaningful temporal traces (Fig. 4D) that exhibit state-specific patterns. Interestingly, the 3-rd BB consistently shows minimal activity across all states, suggesting it captures background or noise activity. The structure of the identified BBs exhibits subtle yet significant adaptations across states in terms of neuron weights and BB assignments. Furthermore, SiBBlInGS finds neurons belonging to multiple neural clusters, suggesting their involvement in multiple functions. When examining the temporal correlations of corresponding BBs within and between states, all BBs exhibited a within/between ratio significantly $> 1$ (Fig. 4E, Fig. 12, Sec. J.4). This indicates reduced variability within states and clear distinctions between states. Furthermore, multi-class logistic regression based only on the identified temporal traces, was able to accurately predict states (Fig. 4C).

**SiBBlInGS discovers emerging local BBs preceding epileptic seizures:** Finally, we applied SiBBlInGS to EEG recordings of an epileptic patient from Handa et al. (2021); Nasreddine (2021) (refer to Sec. K for application details). Specifically, we examined data from an 8-year-old individual who had experienced 5 complex partial seizures (CPS) localized around electrode F8. SiBBlInGS unveiled interpretable and localized EEG activity in the period preceding seizures (Fig. 5), a feat not achieved by other methods (Fig. 13). In particular, it identified a BB specific to the region around the clinically labeled area (F8) that emerged during the 8 seconds prior to the seizure (Fig. 5, middle, prominent pink circle). Additionally, several alterations in BB composition were evident during the seizure in comparison to the normal activity period (Fig. 5, right vs. left panels). For example, the contribution of T4 to the red BB during normal activity is higher than its contribution during a seizure, while the contribution of T5 to the same BB is larger during a seizure. This example underscores the potential of SiBBlInGS in discovering BBs that uniquely emerge under specific states, made possible by the flexibility of $\nu$ to support both state-variant and state-invariant BBs.

## 6 CONCLUSION

We propose SiBBlInGS for graphs-driven identification of interpretable cross-state BBs with their temporal profiles in multi-way time-series data—thus provides insights into system structure and variability. Unlike other approaches, SiBBlInGS naturally accommodates variations in trial numbers, durations, sampling rates, BB sensitivity to states, and subtle changes in cross-states BB structures. We demonstrate SiBBlInGS' capacity to identify functional neural assemblies and discern cross-state variations in web-search data structures, showcasing its promise in additional domains, including, e.g., detecting gene expression clusters in health vs disease, unveiling financial patterns across locations based on stock data, and studying activity shifts in social media dynamics. Regarding limitations, SiBBlInGS assumes Gaussian data statistics, yet Poisson may be more suitable in certain cases. Also, exploring advanced distance metrics for graph construction holds promise for future research. Additionally, the identified BBs currently do not consider potential directed connectivity within them, presenting an exciting opportunity for future research.

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

SUPPLEMENTARY MATERIAL

# A  NOTATIONS

Throughout the paper, we adopt the following notations: superscript $()^d$ refers to state $d$, and subscript $()_m$ refers to trial $m$. Specifically, $\boldsymbol{Y}_m^d$ and $\Phi_m^d$ denote the observations and temporal traces of trial $m$ of state $d$, while $\boldsymbol{A}^d$ represents the BBs of state $d$. Additionally, for a general matrix $\boldsymbol{Z}$, let $\boldsymbol{Z}_{i:}$ ($\boldsymbol{Z}_{:j}$) denote the $i$-th row ($j$-th column) of a matrix $\boldsymbol{Z}$.

Table 1: List of notations for SiBBLINGS.

| Symbol | Description |
|---|---|
| BBs | Building Blocks |
| channels | Each feature in the observations, e.g., neurons in recordings |
| states | Different "views" of the observations. e.g. different cognitive tasks |
| trials/sessions | Repeated observations within state |
| $p$ | Number of BBs |
| $D$ | Number of states |
| $M_d$ | Number of trials for state $d$ |
| $N$ | Number of channels |
| $\boldsymbol{Z}_{n:}$ (or $\boldsymbol{Z}_{[n,:]}$) | The $n$-th row of a general matrix $\boldsymbol{Z}$ |
| $\boldsymbol{Z}_{:i}$ (or $\boldsymbol{Z}_{[:,i]}$) | The $i$-th column of a general matrix $\boldsymbol{Z}$ |
| $L_d$ | Label of state $d$ (optional, can be a scalar or a vector) |
| $\boldsymbol{Y}_m^d \in \mathbb{R}^{N \times T_m^d}$ | Observation for trial $m$ and state $d$ |
| $\boldsymbol{A}^d \in \mathbb{R}^{N \times p}$ | Matrix of BBs for state $d$. |
| $\Phi_m^d \in \mathbb{R}^{N \times T_m^d}$ | Matrix of temporal traces for trial $m$ of state $d$. |
| $\boldsymbol{P} \in \mathbb{R}^{D \times D}$ | States similarity graph |
| $\boldsymbol{H} \in \mathbb{R}^{N \times N \times D}$ | Channel similarity graph |
| $\boldsymbol{\nu} \in \mathbb{R}^p$ | Controls the relative level of cross-state similarity for each BB |
| $\boldsymbol{V} = diag(\boldsymbol{\nu})$ | A diagonal matrix whose entry in index $ii$ is the $i$-th entry of $\boldsymbol{\nu}$ |
| $\beta$ | An hyperparameter controlling the strength of regularization |
| $\gamma_1, \gamma_2, \gamma_3, \gamma_4$ | Hyperparameters to regularize $\boldsymbol{\Phi}_m^d$ |
| $\sigma_{\widetilde{\boldsymbol{H}}}, \sigma_{\boldsymbol{P}}$ | Hyperparameters that control the bandwidth of the kernel |
| $\boldsymbol{\psi}_n^{ij} \in \mathbb{R}^{M_j, M_i}$ | Transformation of the data from state $i$ to state $j$ for channel $n$ |

---

**Algorithm 1** SiBBlInGS Model Training (short version)

---

  **Inputs**
  $\{\boldsymbol{Y}_m\}_{m=1:M_1}^1, ...\{\boldsymbol{Y}_m\}_{m=1:M_D}^D$      $\triangleright$ Observations under $D$ states, $M_d$ trials for state $d$
  $\{\beta, \xi, \epsilon, \gamma_1, \gamma_2, \gamma_3, \gamma_4, \nu, K, w_{graph}, \sigma_p, \sigma_H\}$, $L$ (optional labels)      $\triangleright$ Model parameters
  **Initialization and pre-Calculations**
  $\boldsymbol{A}^d, \{\boldsymbol{\Phi}_m^d\}_{m=1:M_d}$    $\forall d = 1 \ldots D$      $\triangleright$ Initialize BBs and temporal matrices
  $\boldsymbol{P} \in \mathbb{R}^{D \times D}, \boldsymbol{H} \in \mathbb{R}^{N \times N \times D}$      $\triangleright$ Calculate similarity graphs
  **while** not all states converged **do**      $\triangleright$ Repeat until convergence of all states
    **for all** $d = 1 \ldots D$ **do**      $\triangleright$ Iterate over states
      Select a random batch of trials from state $d$      $\triangleright$ Take a batch
      Update $\boldsymbol{A}^d$ and $\boldsymbol{\lambda}^d$      $\triangleright$ via equation 1
      **for all** $m = 1 \ldots M_d$ **do**      $\triangleright$ for every trial in the state
        update $\boldsymbol{\Phi}_m^d$      $\triangleright$ via equation 2

---

# B  FURTHER OPTIONS FOR $\boldsymbol{P}$ COMPUTATION

Here, we explore additional approaches for computing the state-similarity graph $\boldsymbol{P}$. These options take into account factors like data properties, single vs. multi-trial cases, variations in trial duration, and the desired approach (supervised or data-driven).

Table 2: Assumptions and capabilities comparison between SiBBlInGS and other methods.

| Method | SiBBlInGS | mTDR | PCA | Fast ICA | NMF | GPFA | SRM | HOSVD | PARAFAC |
|---|---|---|---|---|---|---|---|---|---|
| Do not force orthogonality? | V | X | X | V | V | V | X | X | V |
| Sparse? | V | X | X | X | X | X | X | X | X |
| Flexible in time across states? | V | X | X | X | X | X | X | V | V |
| Support variations in BB across states? | V | X | X | X | X | X | X | X | X |
| Used for condition variability? | V | V | X | X | X | X | X | V | V |
| Works on tensors? | V | V | X | X | X | X | V | V | V |
| Consider both within & between states variability? | V | V | na | na | na | na | X | X | X |
| Supports state-specific emerging components? | V | V | na | na | na | na | V | X | X |
| Works on non-consistent data duration or sampling rates? | V | X | na | na | na | na | X | X | X |
| Can prior knowledge (labels) control state similarity? | V | V | na | na | na | na | V | X | X |
| Ability to define both state-specific and background components? | V | V | na | na | na | na | X | X | X |
| Supports non-negative decomposition? | V | X | X | X | V | X | X | X | V |

## B.1 SUPERVISED $P$

### B.1.1 CATEGORICAL OR SIMILAR-DISTANCED STATES

For cases where observation states are represented by categorical labels, and we expect a high degree of similarity between all possible pairs of states (i.e., no pair of labels is closer to each other than to another pair), we can define the state similarity matrix $P$ to be identical for all pairs of states. $P$ is then constructed as

$$P = \mathbf{1} \otimes \mathbf{1}^T + c\mathbf{I}, \qquad (3)$$

where $P = \mathbf{1} \otimes \mathbf{1}^T \in \mathbb{R}^{D \times D}$ is a matrix of all ones, $I \in \mathbb{R}^{D \times D}$ is the identity matrix, and $c$ is a weight that scales the strength of self-similarity with respect to cross-state similarities.

## B.2 DATA-DRIVEN $P$

When prior knowledge about state similarity is uncertain or unavailable, SiBBlInGS also provides an unsupervised, data-driven approach to calculate $P$ based on the distance between data points across states. Here we discuss the four options for constructing the matrix $P$ in a data-driven manner, depending on the structure of the observations.

### B.2.1 SINGLE-TRIAL PER-STATE WITH EQUAL-LENGTH ACROSS STATES:

This case refers to the scenario of a single trial for each state ($M_d = 1 \forall d = 1 \dots D$), where all cross-state trials have the same length ($T_1^d = T \ \forall d = 1 \dots D$). Here, the similarity graph $P$ is constructed as

$$P_{d,d'} = \exp\left(-||\boldsymbol{Y}_1^d - \boldsymbol{Y}_1^{d'}||_F^2 / \sigma_{\boldsymbol{P}}^2\right), \qquad (4)$$

where $\sigma_{\boldsymbol{P}}$ controls the bandwidth of the kernel (more options and information about $P$ reconstruction are found in section B).

### B.2.2 MULTIPLE TRIALS PER STATE, SAME TRIAL DURATION

In the most general case where all trials have the same temporal duration, the similarity matrix $P$ is computed by evaluating the distance between the values of each pair of states, considering all trials within each state. For this, we first find the transformation $\boldsymbol{\psi}_n^{ij} \in \mathbb{R}^{M_j \times M_i}$ between the observations of state $i$ to the observation of state $j$, by solving the Orthogonal Procrustes problem (Golub & Van Loan, 2013; Gower, 2004). For this, let $\boldsymbol{Y}^{i*} \in \mathbb{R}^{M_i \times (T \times N)}$ be the matrix obtained by vertically concatenating the flattened observations from each trial ($m = 1 \dots M_i$) of state $i$. Then, the optimal transformation from the observations of state $i$ ($\boldsymbol{Y}^{i*} \in \mathbb{R}^{M_i \times (T \times N)}$) to the observations of state $j$

$(\boldsymbol{Y}^{j*} \in \mathbb{R}^{M_j \times (T \times N)})$ will be

$$\widehat{\boldsymbol{\psi}}^{ij} = \arg\min_{\boldsymbol{\psi}^{ij}} \|\boldsymbol{\psi}^{ij}\boldsymbol{Y}^{i*} - \boldsymbol{Y}^{j*}\|_F^2, \tag{5}$$

where this mapping projects the multiple trials of state $i$ into the same space as of state $j$, via $\widetilde{\boldsymbol{Y}}^{i*} = \widehat{\boldsymbol{\psi}}^{ij}\boldsymbol{Y}^{i*}$. The state similarity matrix will thus be

$$\boldsymbol{P}_{ij} = \exp\left(-\|\widetilde{\boldsymbol{Y}}^{i*} - \boldsymbol{Y}^{j*}\|_F^2/\sigma_p{}^2\right), \tag{6}$$

for all states $i, j = 1 \ldots D$, and where $\sigma_p$ controls the kernel bandwidth.

### B.2.3   SINGLE-TRIAL PER STATE, DIFFERENT DURATION

Further generalization of the state similarity computation requires addressing the case of trials being of varying duration. When the observations correspond to the same process and their alignment using dynamic time warping is justifiable, we can replace the Gaussian kernel measure with the Dynamic Time Warping (DTW) distance metric (Berndt & Clifford, 1994). When we observe a single trial for each state, the similarity metric becomes the average DTW distances over all channels,

$$\boldsymbol{P}_{ij} = \exp\left(-\frac{1}{N}\sum_{n=1}^{N} DTW(\boldsymbol{Y}_{n:}^i, \boldsymbol{Y}_{n:}^j)\right). \tag{7}$$

### B.2.4   MULTIPLE TRIALS PER STATE, DIFFERENT DURATION

Similarly, for the multi-trial case we have

$$\boldsymbol{P}_{ij} = \exp\left(-\frac{1}{N}\sum_{n=1}^{N}\left(\frac{1}{M_j}\sum_{m=1}^{M_j} DTW\left((\widetilde{\boldsymbol{Y}}^{i*})_{m,[(n-1)T:nT]}, (\boldsymbol{Y}^{j*})_{m,[(n-1)T:nT]}\right)\right)\right), \tag{8}$$

where, as before, $\boldsymbol{Y}^{j*}$ is the stacked-trials version of the observations at state $j$ in channel $n$, such that $(\boldsymbol{Y}^{j*})_{m:}$ is the $m$-th row of this matrix, and $(\boldsymbol{Y}^{j*})_{m,[(n-1)T:nT]}$ corresponds to the $m$-th row of this matrix, but limited to the columns ranging from $(n-1)T$ to $nT$. $\widetilde{\boldsymbol{Y}}^{i*}$, as before, refers to the re-ordered version of $\boldsymbol{Y}^{i*}$ according to $\boldsymbol{Y}^{j*}$. It is crucial to note that this approach operates under the assumption that the trials being compared depict similar processes, and that aligning them using DTW is a valid assumption. By aligning the rows using DTW, we can assess the dissimilarity between the trials while accommodating potential temporal distortions and variations in the time axis.

## C   CHANNEL-SIMILARITY KERNEL ($\boldsymbol{H}$)—GENERATION AND PROCESSING

The kernel post-processing involves several steps. First, we construct the kernel $\widetilde{\boldsymbol{H}}^d$ for each state $d = 1 \ldots D$, as described in Equation (1). To incorporate similarities between each possible pair of states $d' \neq d$, where $d, d' = 1 \ldots D$, we perform a weighted average of each $\boldsymbol{H}^d$ with the kernels of all other states, using $\boldsymbol{P}_{d:}$ for the weights, as it quantifies the similarity between state $d$ and all other states: $\boldsymbol{H}^d = \sum_{d'}^{D} \boldsymbol{P}_{dd'}\widetilde{\boldsymbol{H}}^{d'}$. Then, to promote a more robust algorithm, we only retain the $k$ highest values (i.e., k-Nearest Neighbors; kNN) in each row, while the rest are set to zero. The value of $k$ is a model hyperparameter, and depends on the desired BB size. We then symmetrize each state's kernel by calculating $\boldsymbol{H}^d \leftarrow \frac{1}{2}\left(\boldsymbol{H}^d + (\boldsymbol{H}^d)^T\right)$ for all $d = 1 \ldots D$. Finally, the kernel is row-normalized so that each row sums to one, as follows: Let $\Lambda^d$ be a diagonal matrix with elements representing the row sums of $\boldsymbol{H}^d$, i.e., $\Lambda_{ii}^d = \text{diag}\left(\sum_{n=1}^{N}\boldsymbol{H}_{i,n}^d\right)$. The final normalized channel similarity kernel is obtained as $\boldsymbol{H}_{\text{final}}^d = (\Lambda^d)^{-1}\boldsymbol{H}^d$.

## D   SOLVING $\boldsymbol{\Phi}$ IN PRACTICE

In Section 4, the model updates the temporal traces dictionary $\boldsymbol{\phi} = \boldsymbol{\Phi}_m^d$ for all $m = 1 \ldots M_d$, $d = 1 \ldots D$ using an extended least squares for each time point $t$, i.e.,

$$\widetilde{\boldsymbol{\phi}}_{[t,:]} = \arg\min_{\boldsymbol{\phi}_{[t,:]}} \|\widetilde{\boldsymbol{Y}}_{m_{[:,t]}}^d - \widetilde{\boldsymbol{M}}\boldsymbol{\phi}_{[t,:]}\|_2^2, \tag{9}$$

where $\phi_{[t,:]} \in \mathbb{R}^p$ is the dictionary at time $t$,

$$\widetilde{\boldsymbol{Y}}_{m_{[:,t]}}^d = \begin{bmatrix} \boldsymbol{Y}_{m_{[:,t]}}^d \\ [\boldsymbol{0}]_{p \times 1} \\ \gamma_2 \boldsymbol{\phi}_{[t,:]}^{(iter-1)} + \gamma_3 \boldsymbol{\phi}_{[(t-1),:]} \end{bmatrix}, \quad \text{and} \quad \widetilde{\boldsymbol{M}} = \begin{bmatrix} \boldsymbol{A}^d \\ \gamma_4([1]_{p \times p}\sqrt{\boldsymbol{D}} - (\boldsymbol{I}_{p \times p} \circ \sqrt{D})) \\ (\gamma_1 + \gamma_2 + \gamma_3)\boldsymbol{I}_{p \times p} \end{bmatrix},$$

with all parameters being the same as those defined in Section 4 of the main text. Here, $[0]_{p \times 1} \in \mathbb{R}^p$ represents a column vector of zeros, $[1]_{p \times p}$ represents a square matrix of ones with dimensions $p \times p$, and $\boldsymbol{Y}_{m_{[:,t]}}^d \in \mathbb{R}^N$ denotes the measurement in the $m$-th trial of state $d$ at time $t$.

## E    MODEL COMPLEXITY

SiBBIInGS relies on 4 main computational steps:

**Channel Graph Construction:** This operation, performed once for all $N$ channels of every state $d = 1 \ldots D$, generates a channel graph $\boldsymbol{H} \in \mathbb{R}^{N \times N}$ for each state $d \in [1, D]$ by concatenating within-state trials $1 \ldots M_d$ horizontally, resulting in a $N \times \sum_m^{M_d} T_m^d$ matrix. For simplicity, let $\widetilde{T} = \sum_m^{M_d} T_m^d$. The computation complexity of calculating the pairwise similarities of this concatenated matrix for all $D$ states is thus $\mathcal{O}\left(D\tilde{T}^2 N(N-1)\right)$.

For the k-threshold step (B.2.1), that involves keeping only the $k$ largest values in each row while setting the other values to zero—the complexity will be $\mathcal{O}(\tilde{(T)} \log k)$ per row for a total computational complexity of $\mathcal{O}(DN(\tilde{T}) \log k)$ for $N$ rows and $D$ states.

**State Graph Construction:** This is a one-time operation that involves calculating the pairwise similarities between each pair of states. For simplicity, if we assume the case of user-defined scalar labels, and as in this case there are $D$ states (and accordingly $D$ labels), the computation includes $D(D-1))/2$ pairwise distances for $\mathcal{O}\left(D^2\right)$.

**BB Inference ( equation 1):** This iterative step involves per-channel re-weighted $\ell_1$ minimization. If the computational complexity of a weighted $\ell_1$ is denoted as $\mathcal{C}$, then the computational complexity of the Re-Weighted $\ell_1$ Graph Filtering is $NL\mathcal{C} + LNk$, where $N$ is the number of channels, $L$ is the number of iterations for the RWLF procedure, and $k$ is the number of nearest neighbors in the graph. For the last term in equation 1, there are $p^2$ multiplicative operations involving the vector $\nu$ and the difference in BBs, with the exponent $^2$ arising from the $\ell_2^2$ norm. Additionally, there is an additional multiplication step involving the scalar $\boldsymbol{P}_{dd'}$. For each state $d$, this calculation repeats itself $D-1$ times, corresponding to all states that are different from that $d$ state. This process is carried out for every $d = 1 \ldots D$. In total, these multiplicative operations sum up to $\left(p^2 + 1\right)D(D-1)$, resulting in a computational complexity of $\mathcal{O}\left(D^2 p^2\right)$.

**Optimization for $\phi$:** This step refers to the least-squares problem presented in equation 9. If a non-negative constraint is applied, SiBBIInGSuses scipy's nnls for solving $\widetilde{\phi}_{[t,:]} = \arg\min_{\phi_{[t,:]}} \|\widetilde{\boldsymbol{Y}}_{m_{[:,t]}}^d - \widetilde{\boldsymbol{M}}\phi_{[t,:]}\|_2^2$, where $\boldsymbol{Y}^d \in R^{(N+2p) \times T_d}$ and $\boldsymbol{M} \in R^{(N+2p) \times p}$. This results in complexity of $\mathcal{O}\left(p(N+2p)^2 FT\right)$, for all $T$ time points, where $F$ is the number of nnls iterations. Without non-negativity constraint, this problem is a least squares problem with a complexity of $\mathcal{O}\left(Tp^2(2p+N))\right)$. Potential complexity reduction options include: parallelizing RWL1 optimizations per channel, using efficient kNN or approximate kNN search for constructing kNN graphs instead of full graphs, and employing dimensionality reduction techniques to expedite nearest neighbor searches.

## F    DATA AND CODE AVAILABILITY

The code used in this study will be shared on GitHub upon publication, ensuring reproducible results. The data used in this study are publicly available and cited within the paper.

# G  GENERAL EXPERIMENTAL DETAILS

All experiments and code were developed and executed using Python version 3.10.4 and are compatible with standard desktop machines.

# H  SYNTHETIC DATA—ADDITIONAL INFORMATION

## H.1  SYNTHETIC GENERATION DETAILS

We initiated the synthetic data generation process by setting the number of channels to $N = 100$ and the maximum number of BBs to $p = 10$. We further defined the number of states as $D = 3$ and determined the number of time points in each observation to be $T^d = 300$, where $d$ represents the state index (here $d \in \{1, 2, 3\}$). We defined the number of trials for each state as one, i.e., $M_d = 1$ for $d = 1, 2, 3$.

We first initialized a "general" BB matrix ($\boldsymbol{A}$) as the initial structure, which will later undergo minor modifications for each state. We determined the number of non-zero values $k^j$ (i.e., the cardinatlity) for each $j$-th BB in the general $\boldsymbol{A}$ matrix by sampling from a uniform distribution between 1 and 21. Next, we sampled the values in each $j$-th BB from normal distribution (zero mean and unit variance), and set all but the top $k_j$ values to zero. For each state $d$, we generated the time-traces $\boldsymbol{\Phi}^d$ via a linear combination of 15 trigonometric signals, such that the temporal trace of the $j$-th BB is defined as $\boldsymbol{\Phi}^d_{:j} = \sum_{i=1}^{15} c_i f_i(\text{freq}_i * x)$ where $x$ is an array of $T = 300$ time points ($x = 1 \dots 300$), $\text{freq}_i$ is an array of random frequencies sampled uniformly on $[0, 5]$, $f$ refers to a random choice between the sine and cosine functions (with probability 1/2 for each), and the sign ($c_i$) was flipped ($+1$ or $-1$) with a probability of 1/2.

During the data generation process, we incorporated checks and updates to $\boldsymbol{A}$ and $\boldsymbol{\Phi}$ to ensure that the BBs and their corresponding time traces are neither overly correlated nor orthogonal, are not a simple function of the states labels, and that different BBs exhibit comparable levels of contributions. This iterative process involving the checks persisted until no further modifications were required.

The first check aimed to ensure that the temporal traces of at least two BBs across all states were not strongly correlated with the state label vector ($[1, 2, 3]$) at each time point. Specifically, we examined whether the temporal traces of a $j$-th BB across all states ($\Phi^1_{tj}, \Phi^2_{tj}, \Phi^3_{tj}$) exhibited high correlation with the state label vector at each time point. This check was important to avoid an oversimplification of the problem by preventing the temporal traces from being solely influenced by the state labels. To perform this check, we calculated the average correlation between the temporal traces and the state labels ($[1, 2, 3]$) at each time point. If the average correlation over time exceeded a predetermined threshold of 0.6, we introduced additional variability in the time traces of the BB that exhibited a high correlation with the labels. This was achieved by adding five randomly generated trigonometric functions to the corresponding BB. These additional functions were generated in the same manner as the original data (with $\boldsymbol{\Phi}^d_{:j} = \sum_{i=1}^{5} c_i f_i(\text{freq}_i \cdot x)$).

The second check ensured that the time traces were not highly correlated with each other and effectively represented separate functions. If the correlation coefficient between any pair of temporal traces of different BBs within the same state exceeded a threshold of $\rho = 0.6$, the correlated traces were perturbed by adding zero-mean Gaussian noise with a standard deviation of $\sigma = 0.02$.

Next, we ensured that the BBs represented distinct components by verifying that they were not highly correlated with each other. Specifically, if the correlation coefficient between a pair of BBs ($\boldsymbol{A}_{:j}, \boldsymbol{A}_{:i}$ for $j, i = 1 \dots 10$) within a state exceeded the threshold $\rho = 0.6$, each BB in the highly-correlated pair was randomly permuted to ensure their distinctiveness.

To prevent any hierarchical distinction or disparity in BB contributions and differentiate our approach from order-based methods like PCA or SVD, we evaluated each BB's contribution by measuring the increase in error when exclusively using that BB for reconstruction. Specifically, for the $j$-th BB of state $d$, we calculated its contribution as $\text{contribution}_j = -\|\widehat{\boldsymbol{Y}}^d - \boldsymbol{A}^d_{:j} \otimes \boldsymbol{\Phi}^d_{:j}\|_F$, where $\otimes$ denotes the outer product. Then, we compared the contributions between every pair of BBs within the same state. If the contribution difference between any pair of BBs exceeded a predetermined

threshold of $10$, both BBs in the pair were perturbed with random normal noise. Subsequently, a hard-thresholding operation was applied to ensure that the desired cardinality was maintained.

To introduce slight variability in the BBs' structure across states, the general basis matrix $\boldsymbol{A}$ underwent modifications for each of the states. In each state and for each BB, a random selection of 0 to 2 non-zero elements from the corresponding BB in the original $\boldsymbol{A}$ matrix were set to zero, effectively introducing missing channels as differences between states, such that $\boldsymbol{A}^d$ is the updated $\boldsymbol{A}$ modified for state $d$. Finally, the data was reconstructed using $\boldsymbol{Y}^d = \boldsymbol{A}^d(\boldsymbol{\Phi}^d)^T$ for each state $d = 1, 2, 3$.

## H.2 EXPERIMENTAL DETAILS TO THE SYNTHETIC DATA

We applied SiBBLInGS to the synthetic data with $p = 10$ components and a maximum number of $10^3$ iterations, while in practice about $50$ iterations were enough to converge (see Fig. 6). The parameters for the $\lambda$ update in Equation equation 1 were $\epsilon = 0.01$, $\beta = 0.09$, and $w_{\text{graph}} = 1$. For the regularization of $\Phi$ in Equation equation 2, the parameters used were $\gamma_1 = 0.1$, $\gamma_2 = 0.1$, $\gamma_3 = 0$, and $\gamma_4 = 0.0001$. $\nu$ was set to be a vector of ones with length $p = 10$. The number of repeats for the $\boldsymbol{A}$ update within an iteration, for each state, is numreps $= 2$. The number of neighbors used in the channel graph reconstruction ($\boldsymbol{H}^d$) is $k = 25$. The python scikit-learn's (Pedregosa et al., 2011) LASSO solver was used for updating $\boldsymbol{A}$ in each iteration. This synthetic demonstration used the supervised case for building $\boldsymbol{P}$, where $\boldsymbol{P}$ was defined assuming similar similarity levels between each pair of states, by defining $\boldsymbol{P} = \boldsymbol{1} \otimes \boldsymbol{1}^T \in \mathbb{R}^{3 \times 3}$ (the case described in Section B.1.1, with $c = 1$).

## H.3 JACCARD INDEX CALCULATION

In Figure 1C, we computed the Jaccard similarity index between the identified BBs by SiBBlInGS and the ground truth BBs. To obtain this measure, we first rearranged the BBs based on the correlation of their temporal traces with the ground truth traces (since the method is invariant to the order of the BBs). Then, we nullified the 15 lower percentiles of the $\boldsymbol{A}$ matrix, which correspond to values close to zero. Finally, we compared the modified identified BBs to the ground truth BBs using the "jaccard_score" function from the sklearn library (Pedregosa et al., 2011).

## H.4 COMPARISON OF SiBBLInGS RESULTS FOR SYNTHETIC DATA

**Initial Extraction of BBs from each method:** To compare SiBBlInGS with other methods, we employed the following approach. For PCA global, we conducted PCA on the entire dataset after horizontally concatenating the time axis. Subsequently, we employed PCA with sklearn (Pedregosa et al., 2011), specifying the number of Principal Components (PCs) to match the number of BBs allowed by SiBBlInGS ($p = 10$). These PCs were then treated as the BBs. In the case of PCA local, we followed a similar procedure. However, we ran PCA individually for each state. For Tucker and PARAFAC, we utilized the Tensorly library (Kossaifi et al., 2021) with a rank set to $p = 10$ (the number of BBs allowed by SiBBlInGS). We interpreted the BBs as the first factor (factors[0] in Tensorly output), and we considered the temporal traces as the second factor (factors[1] in Tensorly output) while multiplying them by the corresponding weights from the state factor (third factor, factors[2]) to enable cross-state flexibility to these temporal traces. For NONFAT Wang & Zhe (2022b), we utilized the code shared by the authors at Wang & Zhe (2022a). The model was executed with the same parameters as specified in Wang & Zhe (2022b), but with rank set to 10 to align with the desired BBs. The algorithm was trained for 500 epochs across 10 folds. BBs were extracted from the two views of the "$Z_arr$" matrix during the last epoch. For single-state BBs, the first view was reweighted using the weights obtained from the second view of "$Z_arr$" for each state and BB. Temporal traces were then extracted from the "$U_arr$" matrix to calculate the trace of each BB under each state. For NNDTN (discrete-time NN decomposition with nonlinear dynamics, as implemented by Wang & Zhe (2022a)), we employed the "$Uvec$" attribute by concatenating individual components of "$v_{i_n}$" to neurons over states over the number of BBs across all time points. The traces were then obtained by optimizing the BBs' activity to the original tensor.

**post-processing steps applied to the BBs and traces of all methods to align them with the ground truth results:** To assess and compare the results of these alternative methods against the ground truth BBs and traces, we initially normalized the BBs to fit the range of the ground truth

BBs, applied sparsity using hard-thresholding such that the identified BBs from each method will present similar sparsity level to that of the ground truth, and then reordered the BBs to maximuze the correlations of their temporal traces with the ground truth traces. This alignment was necessary since SiBBlInGS is insensitive to the ordering of BBs.

For the correlation comparisons ($\rho(\boldsymbol{A}, \widehat{\boldsymbol{A}})$), we examined the correlation between the BBs, as well as their temporal traces, in comparison to the ground truth. Recognizing that correlation might not be the most suitable metric for sparse BBs comparison, we further evaluated the clustering performance using the Jaccard index as well. However, to utilize the Jaccard index as a metric and considering that the BBs of these methods are inherently dense (not sparse), we introduced artificial sparsity through hard thresholding. To ensure a fairer comparison and align it with SiBBlInGS (which naturally generates sparse BBs), we employed the sparsity level of the ground truth BBs for each state as the hard thresholding parameter for the BBs of the other methods.

# I  GOOGLE TRENDS—ADDITIONAL INFORMATION

## I.1  TRENDS DATA ACQUISITION AND PRE-PROCESSING

The acquisition and pre-processing of Google Trends data involved manually downloading the data from April 1, 2010, to November 27, 2022, for each of the selected states: California (CA), Maryland (MD), Michigan (MI), New York (NY), Illinois (IL), Louisiana (LA), Florida (FL), and Washington (WA), directly from the Google Trends platform (Google Trends, Accessed 11 November 2022). The comprehensive list of terms, as clustered according to SiBBlInGS, is presented in Figure 7. To ensure comprehensive coverage of search patterns, the data was downloaded by examining each query in all capitalization formats, including uppercase, lowercase, and mixed case.

The data (in CSV format) was processed using the 'pandas' library in Python (pandas development team, 2020; Wes McKinney, 2010) and keeping only the relevant information from January 2011 to October 2022, inclusively. We conducted a verification to ensure the absence of NaN (null) values for each term in every selected state. This step confirmed that no terms or states were inadvertently missed during the data downloading process. To pre-process each term, we implemented a two-step normalization procedure. First, the values within the chosen date range were scaled to a maximum value of 100. This step ensured that the magnitude of each term's fluctuations remained within a consistent range. Next, the values for each term were divided by the sum of values across all dates and then multiplied by 100, resulting in an adjusted scale where the area under the curve for each term equaled 100. This normalization procedure accounted for potential variations in the frequency and magnitude of term occurrences, enabling fair comparisons across different terms. By applying these pre-processing steps, we aimed to mitigate the influence of isolated spikes or localized peaks that could distort the overall patterns and trends observed in the data. Since the focus of this processing was on assessing the relative contribution of a term within a BB rather than comparing the overall amplitude and mean of the term across states, factors such as population size and other characteristics of each state were not taken into consideration.

## I.2  EXPERIMENTAL DETAILS FOR GOOGLE TRENDS

We ran the Trends experiment with $p = 5$ BBs, and applied non-negativity constraints to both the BB components and their temporal traces. The $\boldsymbol{\lambda}$'s parameters in Equation equation 1 included $\epsilon = 9.2$, $\beta = 0.01$, and $w_{\text{graph}} = 35$. For the regularization of $\boldsymbol{\Phi}$ in Equation equation 2, we used the parameters $\gamma_1 = 0$, $\gamma_2 = 0$, $\gamma_3 = 0.05$, $\gamma_4 = 0.55$. The trends example used the data-driven version for studying $\boldsymbol{P}$, and we set $\boldsymbol{\nu}$ to be a vector of ones with length $p = 5$.

During each iteration, $\boldsymbol{A}$ underwent two updates within each state. The number of neighbors we used in the channel graph reconstruction was $k = 4$. We used the PyLops package in Python, along with the SPGL1 solver (Ravasi & Vasconcelos, 2020) to update $A$ in each iteration. With respect to SPGL1 parameters (as described in (Ravasi & Vasconcelos, 2020)), we set the initial value of the parameter $\tau$ to 0.12, and a multiplicative decay factor of 0.999 was applied to it at each iteration. We note here that SPGL1 solves a Lagrangian variation of the original Lasso problem, where, i.e., it bounds the $\ell_1$ norm of the selected BB to be smaller than $\tau$, rather than adding the $\ell_1$ regularization to the cost (van den Berg & Friedlander, 2008; Ravasi & Vasconcelos, 2020).

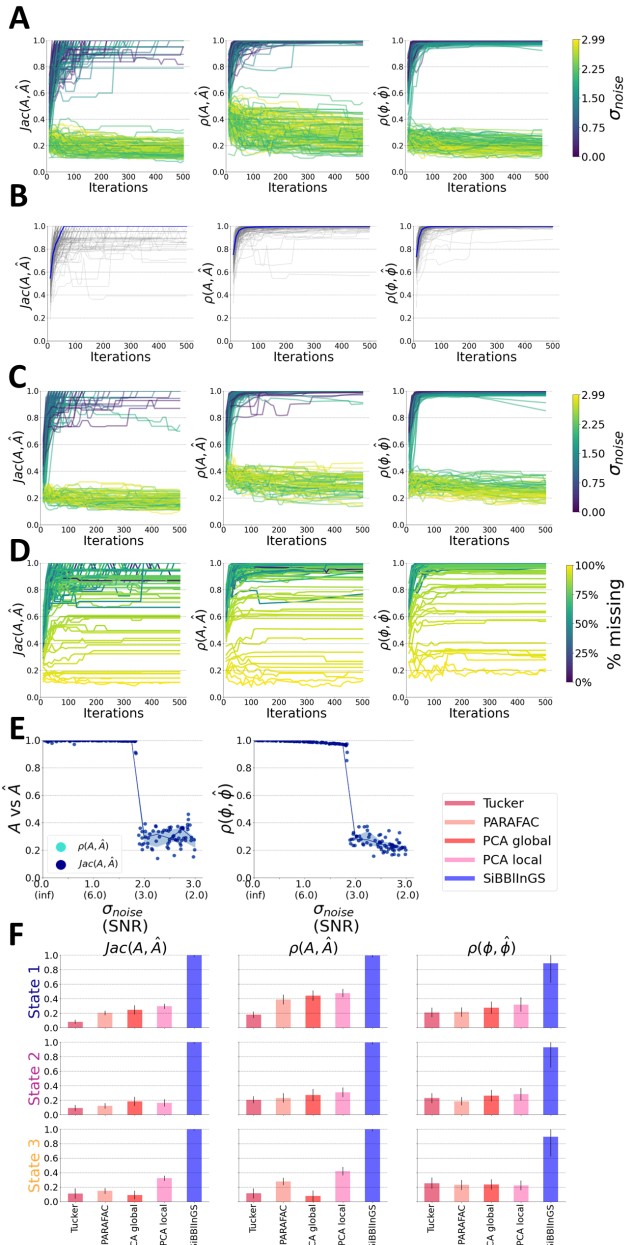

Figure 6: **Synthetic Data Results Robustness - cont. A** Model performance under increasing levels of noise along with random initializations, over the model training iterations. Color: increasing levels of missing samples. Left: Jaccard index between the recovered $A$ and the ground true $A$. Middle: Correlation between the recovered $A$ and the ground true $A$. Right: Correlation between the recovered $\Phi$ and the ground true $\Phi$. **B** Model performance under random initializations (no noise), over the model training iterations. The blue curve is the median over all repeats. **C** Model performance under increasing levels of noise only (fixed initializations). **D** Model performance under increasing levels of missing samples, over the model training iterations. **E** SiBBlInGSPerformance under increasing noise levels. **F** Comparison to other relevant methods, for each state individually (SiBBlInGSin blue, other methods in pink to red colors).

|  | CA | FL | IL | LA | MD | MI | NY | WA |
|---|---|---|---|---|---|---|---|---|
| **BB 1** | Berkeley, Campus, College, Harvard, Phd, Princeton | Admissions, Campus, College, Harvard, Phd, Princeton | Admissions, Campus, College, Harvard, Phd, Princeton | Admissions, Campus, College, Harvard, Phd, Princeton | Admissions, Campus, College, Harvard, Phd, Princeton | Admissions, Campus, College, Harvard, Phd, Princeton | Admissions, Berkeley, Campus, College, Harvard, Princeton | Admissions, Campus, College, Harvard, Phd, Princeton |
| **BB 2** | Afikomen, Chametz, Charoset, Haggadah, Pesach, Seder | Afikomen, Chametz, Charoset, Haggadah, Passover, Pesach | Chametz, Charoset, Haggadah, Passover, Pesach, Seder | Chametz, Charoset, Haggadah, Passover, Pesach, Seder | Afikomen, Chametz, Charoset, Haggadah, Passover, Pesach | Chametz, Charoset, Haggadah, Passover, Pesach, Seder | Afikomen, Chametz, Charoset, Haggadah, Pesach, Seder | Chametz, Charoset, Haggadah, Passover, Pesach, Seder |
| **BB 3** | Auld lang syne, Chicken soup, Decorations, Depression, Gpa, Sweets | Auld lang syne, Champagne, Chicken soup, Depression, Gpa, Sweets | Auld lang syne, Chicken soup, Decorations, Depression, Gpa, Sweets | Auld lang syne, Champagne, Chicken soup, Decorations, Gpa, Sweets | Auld lang syne, Champagne, Chicken soup, Decorations, Gpa, Sweets | Auld lang syne, Champagne, Chicken soup, Decorations, Gpa, Sweets | Auld lang syne, Champagne, Chicken soup, Countdown, Decorations, Sweets | Auld lang syne, Champagne, Chicken soup, Decorations, Depression, Sweets |
| **BB 4** | Elf, Gift, New years eve, Poinsettia, Ugly sweater | Elf, Gift, New years eve, Poinsettia, Ugly sweater | Elf, Gift, New years eve, Poinsettia, Ugly sweater | Elf, Gift, New years eve, Poinsettia, Ugly sweater | Elf, Gift, New years eve, Poinsettia, Ugly sweater | Elf, Gift, New years eve, Poinsettia, Ugly sweater | Elf, Gift, New years eve, Poinsettia, Ugly sweater | Elf, Gift, New years eve, Poinsettia, Ugly sweater |
| **BB 5** | Cdc, Hopkins, Kippur, N95, Quarantine, Zoom | Cdc, Hopkins, Mit, N95, Quarantine, Zoom | Cdc, Hopkins, N95, Quarantine, Zoom | Cdc, Hopkins, Mit, N95, Quarantine, Zoom | Cdc, Hopkins, N95, Quarantine, Zoom | Cdc, Hopkins, N95, Quarantine, Zoom | Cdc, Hopkins, N95, Quarantine, Zoom | Cdc, Hopkins, Mit, N95, Quarantine, Zoom |

Figure 7: **Table of clustered words for the Google Trends experiment**

|  | SiBBlInGS | PCA Local | PCA Global | PARAFAC | Tucker |
|---|---|---|---|---|---|
| **BB 1** | Cdc, Hopkins, N95, Quarantine, Zoom | Hopkins, N95, Quarantine, Zoom | Hopkins, N95, New years eve, Quarantine, Ugly sweater, Zoom | | |
| **BB 2** | Afikomen, Chametz, Charoset, Haggadah, Pesach, Seder | Afikomen, Ball drop, Charoset, Elf, Gift, Haggadah, Memorial, N95, Pesach, Ugly sweater, Zoom | Afikomen, Ball drop, Berkeley, Chametz, Depression, Gpa, Haggadah, Harvard, Memorial, N95, New years eve, Passover, Pesach, Seder, Spirit | Admissions, Afikomen, Berkeley, Cdc, Chametz, Charoset, Decorations, Depression, Gpa, Haggadah, Harvard, Instacart, Labor, Matzo ball, Passover, Pesach, Princeton, Seder, Spirit | Elf, Hopkins, N95, New years eve, Poinsettia, Quarantine, Santa, Ugly sweater, Zoom |
| **BB 3** | Auld lang syne, Chicken soup, Decorations, Depression, Gpa, Sweets | Charoset, Elf, Memorial, New years eve, Ugly sweater | Cdc, Chametz, Charoset, Haggadah, N95, Passover, Pesach, Quarantine, Seder, Zoom | Admissions, Ball drop, Cdc, Countdown, Hopkins, Instacart, N95, New years eve, Quarantine, Zoom | Cdc, Chametz, Charoset, Haggadah, N95, Passover, Pesach, Quarantine, Seder, Zoom |
| **BB 4** | Elf, Gift, New years eve, Poinsettia, Ugly sweater | Cdc, N95, Quarantine, Zoom | Afikomen, Auld lang syne, Ball drop, N95, New years eve, Ugly sweater | Auld lang syne, Champagne, Decorations, Elf, Gift, Hopkins, Labor, Memorial, N95, New years eve, Poinsettia, Santa, Ugly sweater | Campus, Charoset, Elf, Labor, Memorial, New years eve, Spirit, Ugly sweater |
| **BB 5** | Berkeley, Campus, College, Harvard, Phd, Princeton | Afikomen, Charoset, Memorial, Zoom | Charoset, Elf, Labor, New years eve, Ugly sweater | Auld lang syne, Ball drop, N95, New years eve | Auld lang syne, Ball drop, Labor, N95, Memorial, New years eve, Ugly sweater |

Figure 8: **Comparison of The Google Trends Results to Other Methods with 5 BBs for CA:** Comparison to other methods, each applied with 5 Building Blocks (BBs) like SiBBlInGS, yielded less interpretable BBs, particularly for the 'CA' (California) theme. BBs for the Google Trends dataset were obtained following an additional thresholding step applied to each model's factorization results, preserving only the top 90% values from each method's BB matrix. SiBBlInGS discerns theme-specific BBs (e.g., 'Covid' and 'University'), while other methods produce more blended compositions. Empty cells for PARAFAC and Tucker indicate that those BBs remained empty.

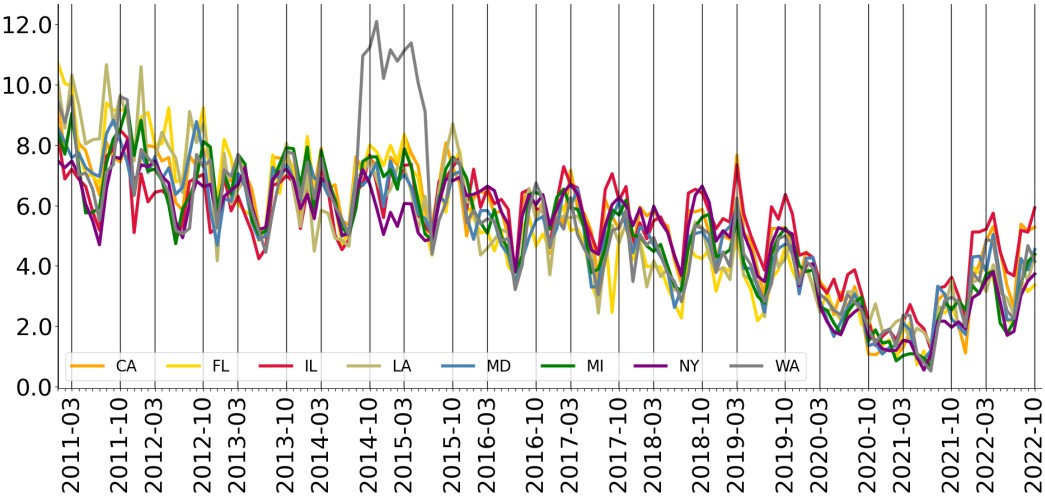

Figure 9: Temporal traces of college admission patterns showing bi-yearly peaks around March and October, aligning with key milestones in the US college admissions process. Additionally, a decrease in online interest in the college BB is observed during the COVID-19 pandemic.

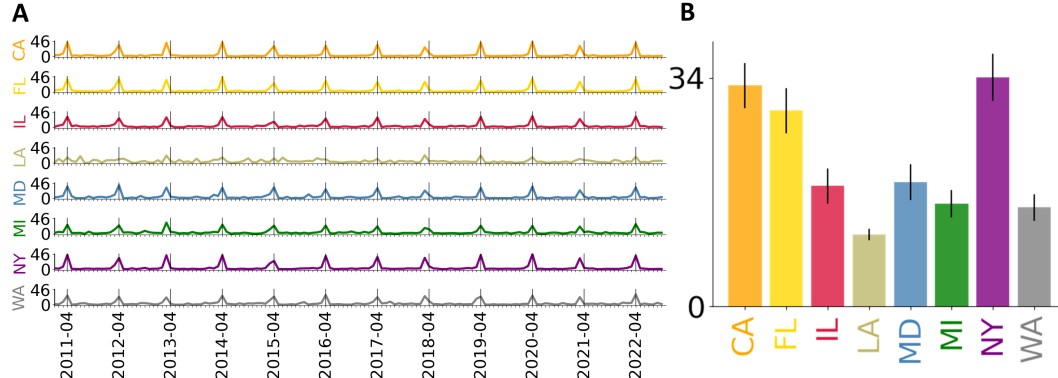

Figure 10: **Temporal trace of Passover BB**. The Passover BB patterns show an alignment with the percentage of Jewish population in different states. **A** Temporal traces of the Passover BB for each state. Vertical black lines indicate the month of April, when Passover is usually celebrated. **B** The mean and standard error of peak values for each state.

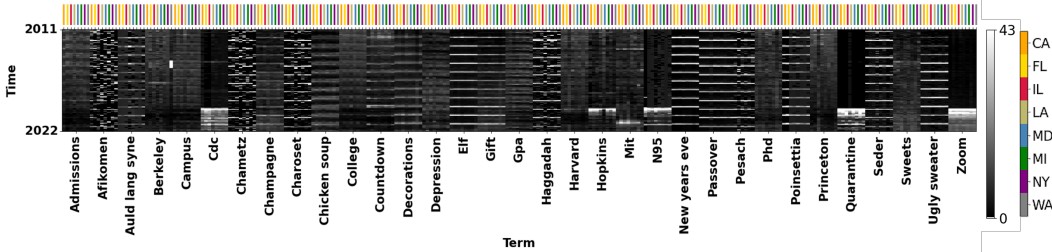

Figure 11: **Post-processed data of the temporal activity of the chosen queries.** Ordered by state and alphabetical order of the queries. The states are markered by the small colorful vertical lines that appear at the top.

### I.3 TEMPORAL TRACES OF COLLEGE BB

The temporal traces of the college admission identified BB exhibit distinct bi-yearly peaks, with notable increases in activity around March and October, along with a clear decrease between March to next October (Fig. 9). These peaks align with key periods in the college admissions cycle, including application submission and admission decision releases. Particularly, around the end of March, many colleges and universities release their regular admission decisions, prompting increased population interest. Similarly, October marks the time when prospective students typically start showing increased interest in applying to colleges as many colleges have early application deadlines that fall in late October or early November. The bi-yearly peaks pattern in March and October thus reflects the concentrated periods of activity and anticipation within the college admissions process. External factors such as the COVID-19 pandemic can also influence the timing and dynamics of the college admissions process, as we observe by the decrease in the college BB activity during the pandemic period (Fig. 9).

### I.4 TEMPORAL TRACES OF PASSOVER BB

SiBBInGS identified a "Passover" BB, characterized by temporal traces that show a clear alignment with the timing of Passover, which usually occurs around April. The time traces demonstrate a prominent peak in states with higher Jewish population percentages, like CA, FL, and NY (Fig. 10), as computed by the average peak value plotted for the different states. The peak finding was done using scipy's (Virtanen et al., 2020) "find_peaks" function with a threshold of 4.

## J NEURAL DATA—ADDITIONAL INFORMATION

### J.1 NEURAL DATA PRE-PROCESSING

In this experiment we used the neural data collected from Brodmann's area 2 of the somatosensory cortex in a monkey performing a reaching-out movement experiment from Chowdhury et al. (Chowdhury & Miller, 2022; Chowdhury et al., 2020). While the original dataset includes data both under perturbed and unperturbed conditions, here, for simplicity, we used only unperturbed trials. We followed the processing instructions provided by Neural Latents Benchmark Pei et al. (2021) to extract the neural information and align the trials. The original neural data consisted of spike indicators per neuron, which were further processed to approximate spike rates by convolving them with a 60-point wide kernel.

For each of the 8 angles, we randomly selected 18 trials, resulting in a total of 144 data matrices. The states were defined as the angles, and for learning the supervised $P$, we used as labels the x-y coordinates of each angle in a circle with a radius of 1 (i.e., sine and cosine projections).

### J.2 EXPERIMENTAL DETAILS FOR THE NEURAL DATA EXPERIMENT

We ran SiBBLInGS on the reaching-out dataset with $p = 4$ BBs. The $\boldsymbol{\lambda}$'s parameters used were $\epsilon = 2.1$, $\beta = 0.03$, and $w_{\text{graph}} = 10.1$. For the regularization of $\boldsymbol{\Phi}$ we used: $\gamma_1 = 0.001$, $\gamma_2 = 0.001$, $\gamma_3 = 0.1$, and $\gamma_4 = 0.3$ and we set $\boldsymbol{\nu}$ to be a vector of length $p = 4$ with $\nu_1 = 0.8$ (to allow more flexibility in the first BB), and $\nu_k = 1$ for $k = 2, 3, 4$. For the neural data, we used the supervised version of $P$, where the $x - y$ coordinates are used as the labels for calculating $P$. During each iteration, $A$ underwent two updates within each state. We chose $k = 7$ neighbors for the channel graph reconstruction, and used Python scikit-learn's (Pedregosa et al., 2011) LASSO solver for the update of $A$.

### J.3 STATE PREDICTION USING TEMPORAL TRACES

We used the identified temporal traces $\boldsymbol{\Phi}$ to predict the state (hand direction). The dimensionality of each state's temporal activity $\boldsymbol{\Phi}^d$ was reduced to a vector of length $p \times 4 = 16$ using PCA with 4 components. A k-fold cross-validation classification approach with $k = 4$ folds was employed. In each iteration, a multi-class logistic regression model with multinomial loss was trained on 3 folds and used to predict the labels of the remaining fold. This process was repeated for each fold, and the results were averaged. The confusion matrix and accuracy scores for each state (angle), as shown in Figure 4C and in Figure 12F.

### J.4 COMPUTATION OF $\rho_{\text{WITHIN/BETWEEN}}$

To compute the correlation for the "within" state case, a random bootstrap approach was employed. Specifically, for each state, we randomly selected 100 combinations of temporal trace pairs corresponding to the same BB but from different random trials within the state, computed the correlations between these temporal trace pairs, and averaged the result over all 100 bootstrapped samples to obtain the average correlation. Similarly, for the "between" states case, we repeated this procedure with the difference that we selected 100 random bootstrapped combinations of pairs of the same BB but from trials of different states. In Figure 12C, the average correlations are shown for each BB. The ratio depicted in Figure 4E represents the ratio between the averages of the "within" and "between" state correlations.

## K EPILEPSY—ADDITIONAL INFORMATION

### K.1 DATA CHARACTERISTICS AND PRE-PROCESSING FOR SiBBLInGS ANALYSIS

The Epilepsy EEG experiment in this paper is based on data kindly shared publicly in (Handa et al., 2021).

The data consist of EEG recordings obtained from six patients diagnosed with focal epilepsy, who were undergoing presurgical evaluation. As part of this evaluation, patients temporarily discontinued

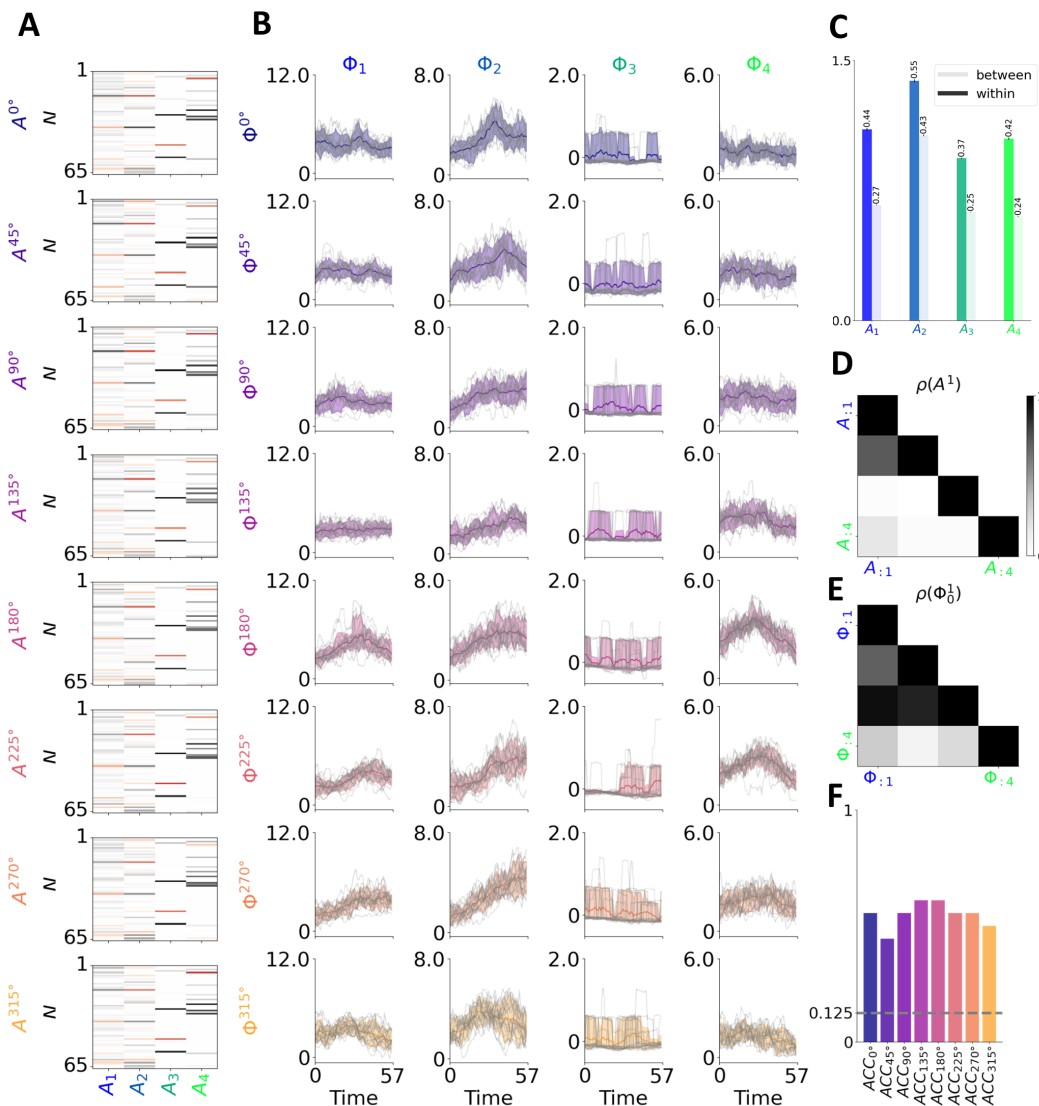

Figure 12: **Additional Figures for the Neural Recordings Experiment**. **A** The identified BBs for the different states. While there is clear consistency, slight modifications can be observed across states, capturing the natural variability in neural ensembles corresponding to different tasks. **B** Temporal traces of the identified $BB$s, shown with a 90% confidence interval (background color), and all trials are plotted in light gray. The color corresponds to the state color used in Figure 4. We observe adaptation over the states as well as differences between the temporal traces of BBs within a given state. The third BB exhibits significantly lower activity compared to the others (see also Figure 4), suggesting that it might capture general background trends or noise. **C** Within and between temporal trace correlations (averaged over 100 bootstrapped examples) with standard error, colored according to the BB color, and transparency representing the strength of the between (opaque) and within (less opaque) correlations. **D** Example of the correlations between each pair of BBs within the 1-st state ($0°$). This shows that while some BBs are orthogonal, others are not. **E** Example of within-state correlations between each pair of temporal traces of the BBs within the 1-st trial of the 1-st state ($0°$), showing that the temporal traces are neither orthogonal nor overly correlated. **F** Accuracy in predicting the state using only the temporal traces of that state as input (colored by the state color). While the random accuracy would be $1/\text{length(labels)} = \frac{1}{8} = 0.125$, the achieved accuracies are significantly higher for all states.

their anti-seizure medications to facilitate the recording of habitual seizures. The data collection period spanned from January 2014 to July 2015.

The EEG data, as described by (Handa et al., 2021), were recorded using a standard 21 scalp electrodes setuc, following the 10-20 electrode system, with signals sampled at a rate of 500 Hz. To enhance data quality, all channels underwent bandpass filtering, with a frequency range from 1/1.6 Hz to 70 Hz. Furthermore. Certain channels, including Cz and Pz, were excluded from some recordings due to artifact constraints. These seizures manifest different patients, seizure types, ictal onset zones, and durations.

Here, we focused on the EEG data from an 8-year-old male patient. This patient experienced five recorded complex partial seizures (CPS) in the vicinity of electrode F8. The EEG data for this patient includes both an interictal segment during which no seizures are recorded and 5 ictal segments representing seizures. ,

To prepare the data for compatibility with the input structure of SiBBlInGS, we divided the epileptic seizure data into non-overlapping batches, with a maximum of 8 batches extracted from each seizure. Each batch had a duration of 2000 time points, equivalent to 4 seconds. This process resulted in 4 seizures with 8 batches each and one seizure with 7 batches due to its shorter duration.

For each seizure, we also included data from the 8 seconds preceding the marked identification of the seizure, as indicated in the data. This amounted to 2 additional 2000-long batches (each corresponding to 4 seconds) before each seizure event.

Regarding normal activity data, we randomly selected 40 batches, each spanning 4 seconds (2000 time points), from various time intervals that did not overlap with any seizure activity or the 8-second pre-seizure period.

In total, we had 40 batches of normal activity, 39 batches of seizure activity, and 10 batches of pre-seizure data.

We ran SiBBlInGS on this data with $p = 7$ BBs. For the state-similarity graph ($P$), we adopted a supervised approach to distinguish between seizure and non-seizure states, as detailed in the categorical case in B.1.1, where we assigned a strong similarity value constraint to same-state trials and lower similarity values to different-state trials.

We also leverage this example to underscore the significance of the parameter $\nu$ in the model's ability to discover networks that emerge specifically under certain states as opposed to background networks. In this context, we defined here $\nu = [1, 1, 1, 1, 1, 1, 0]$ such that the similarity levels of the 1st to 6th BBs are determined by the relevant values in $P$, while the last network's similarity is allowed to vary between states.

During the training of SiBBlInGS on this data, we adopted a training strategy where 8 random batches were selected in each iteration to ensure that the model was exposed to an equal number of trials from each state during each iteration and enhancing its robustness.

## K.2 COMPARING EEG RESULTS TO EXISTING APPROACHES

We extended tensor and matrix factorization methods, including Tucker decomposition, PARAFAC, global PCA, and local PCA, to adapt them for capturing state information and generating sparse clusters in EEG data. Tucker and PARAFAC were implemented using the Tensorly Python package (Kossaifi et al., 2016)(tensorly.decomposition.parafac, tensorly.decomposition.tucker), while PCA was applied using scikit-learn.

For 2D methods (PCA local and PCA global), we applied global PCA to horizontally concatenated data (19 channels x $\sum_i T_i$) with the number of principal components equal to the number of Building Blocks (BBs) used in SiBBlInGS (6). For PCA local, we conducted individual PCA for each state's trials, resulting in $k = 3$ distinct BB matrices.

To address differing state durations in tensor factorization, we horizontally concatenated trials within the same state, zero-padded them to match the longest duration, and created a tensor (N electrodes x time-padded x 3 states) for applying tensor factorization methods.

After applying the aforementioned approaches to the data, we extracted the BBs ($A$). In the cases of global and local PCA, these BBs were treated as the Principal Components (PCs). In the PARAFAC and Tucker tensor-factorization methods, they were considered the first factor (factors[0] from the tensorly output), weighted by the relevant components from the third factor (the states axis, factors[2]).

We then performed the following steps: 1) Normalized the matrices to ensure that each BB had a similar absolute sum of its columns, resulting in BBs of comparable magnitudes for state comparison, and 2) Introduced artificial sparsity into the matrices through hard thresholding, aiming to achieve the same level of sparsity observed in SiBBlInGS for each state. Figure 13 illustrates the outcomes, demonstrating that these approaches failed to detect the emergence of BBs around electrode F8, resulting in widespread non-specific clusters.

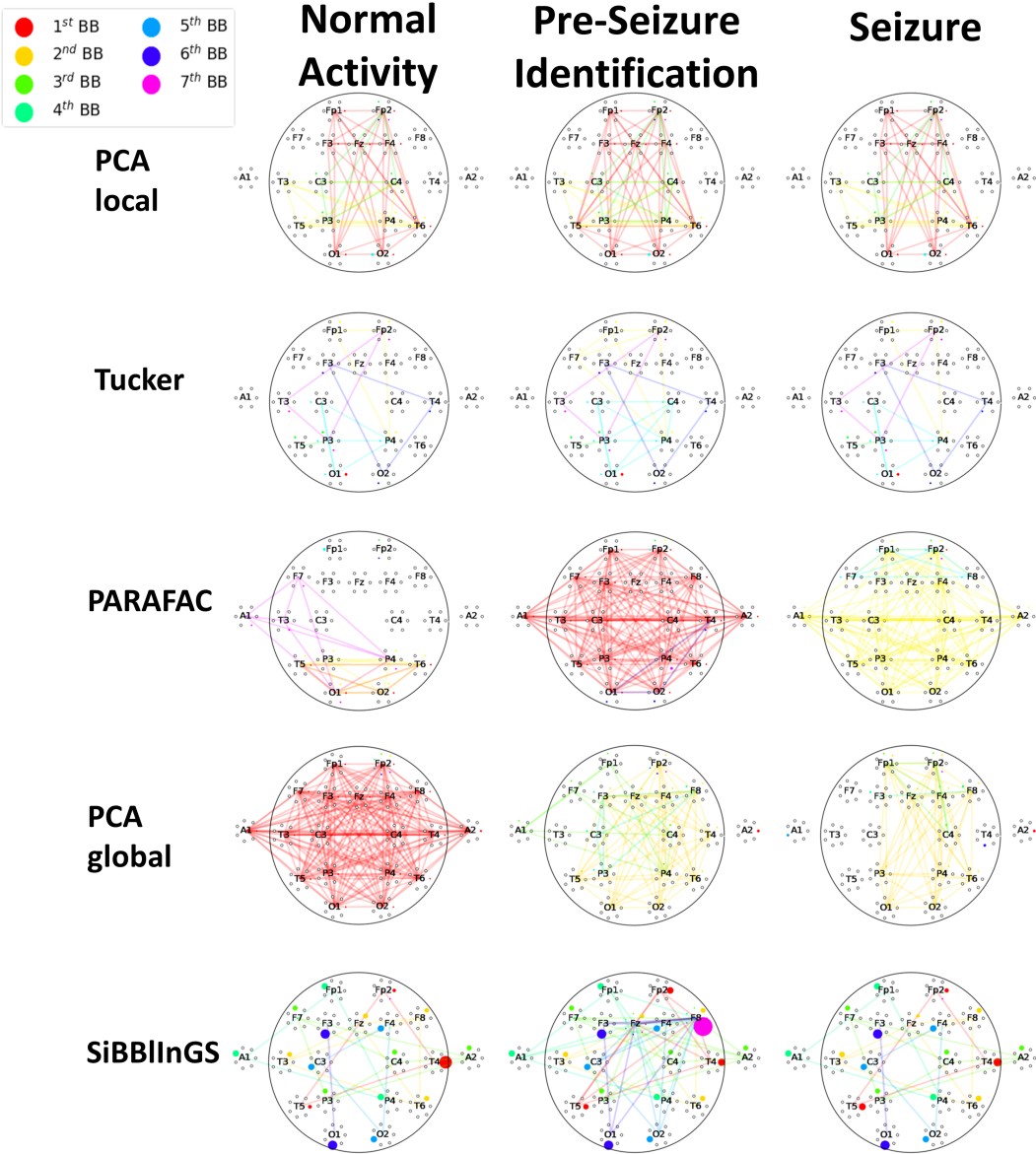

Figure 13: **Comparison of the EEG results to Other Methods:** Various approaches (different rows) failed to identify the BB emerging before the seizure around F8, resulting in widespread uninterpretable networks.

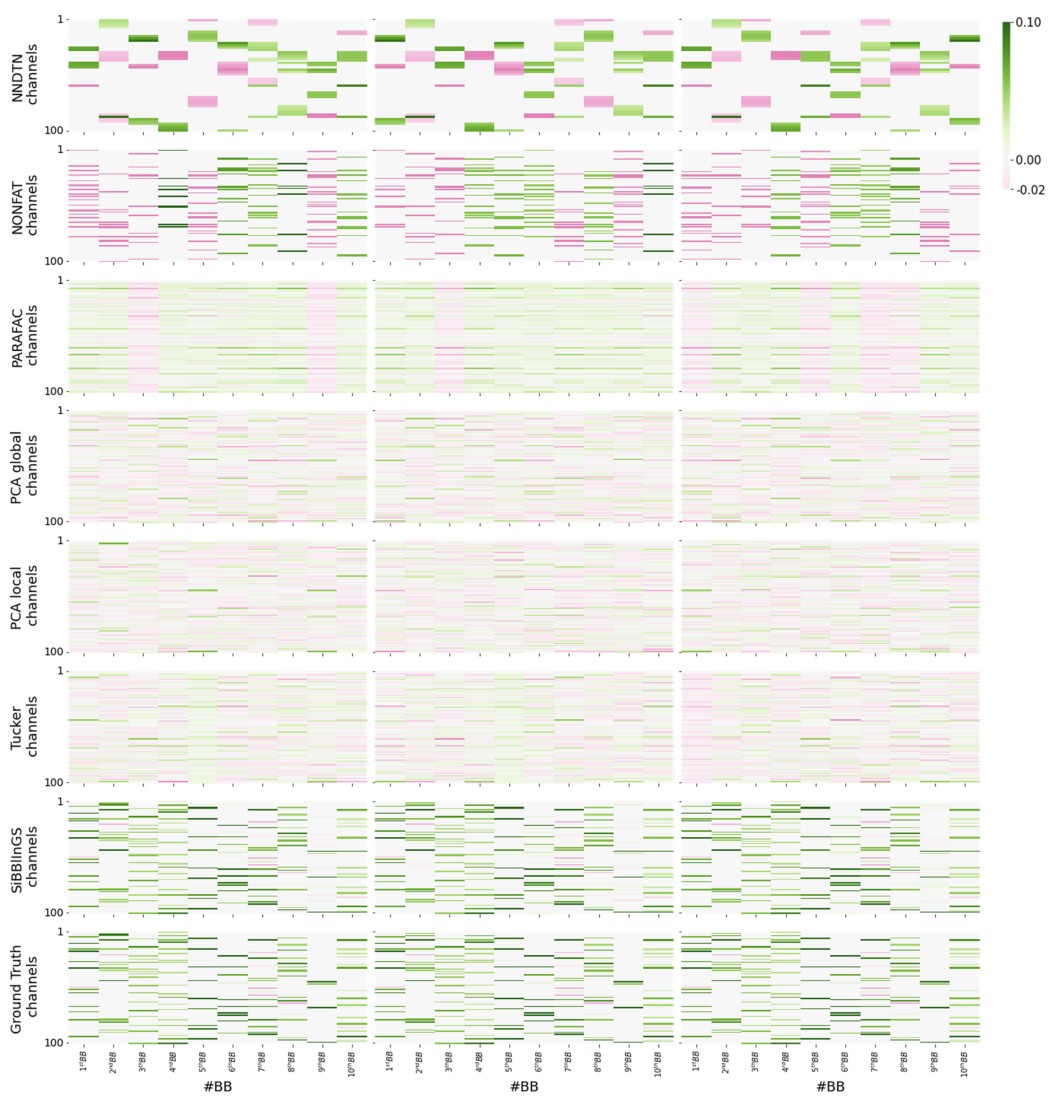

Figure 14: **BBs identified by different methods**. BBs identified by SiBBlInGS are compared with those from other methods, including PARAFAC, Tucker, PCA (global and local), and Gaussian-process-based methods. The identified BBs were reordered to best match the ground truth BBs' temporal traces through maximum correlations. A subsequent hard-thresholding step was applied to achieve sparsity, aligning with the sparsity level with of the ground truth components. The BBs were normalized to sum to 1 each for visualization purposes only.

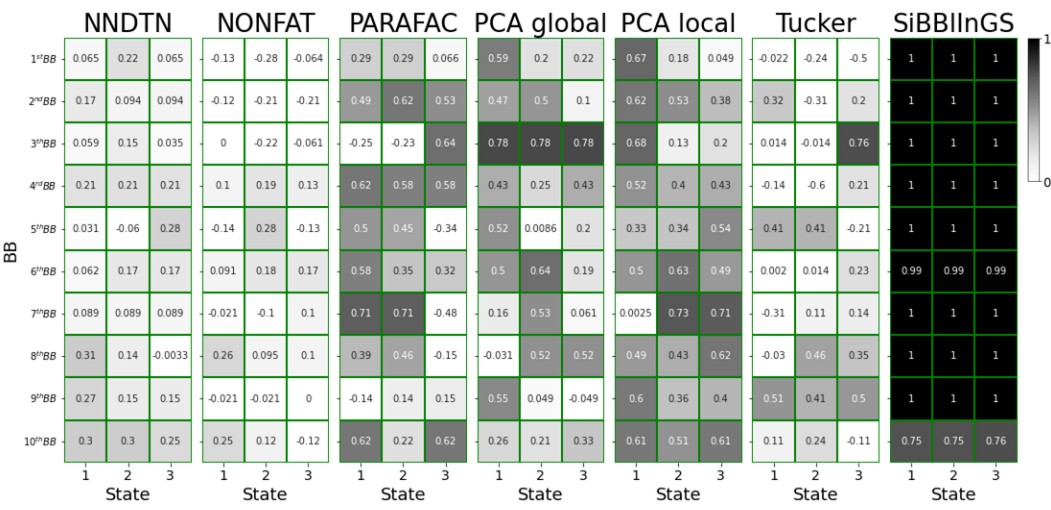

Figure 15: **Correlations between BBs identified by different methods and ground truth BBs for each state and BB number**.

