# OpenReview forum: "SiBBlInGS: Similarity-driven Building Block Inference using Graphs across States"
_ICLR.cc/2024/Conference — Submitted to ICLR 2024_

### Official Review · Reviewer_WRaP · 2023-11-01

**Soundness:** 3 good
**Presentation:** 3 good
**Contribution:** 2 fair
**Rating:** 6
**Confidence:** 3

**Summary:**

The document introduces SiBBlInGS, a framework designed to identify Building Blocks (BBs) and their temporal profiles within high-dimensional, multi-state time-series data. SiBBlInGS utilizes channel-similarity and state-similarity graphs to uncover interpretable BBs, providing insights into the system's structure and variability across different states. The framework is demonstrated to be applicable across various data modalities, offering a deeper understanding of functional circuits, task encoding, and state modulations. It is validated using neural data from a monkey's somatosensory cortex during a reaching-out movement experiment, showcasing its potential in neuroscience for analyzing complex datasets.

**Strengths:**

1. A well-written and structured paper.
2. The proposed SiBBlInG has the ability to account for variations in temporal activities across trials and subtle differences in the composition of Building Blocks (BBs) across states.
3. The SiBBlInGS framework offers valuable insights into functional circuits, task encoding, and state modulations across various data modalities. It is versatile and can be applied across diverse fields, including neuroscience, social science, and genetics.
4. The experimental part is solid and abundant.

**Weaknesses:**

The novelty of the paper mainly comes from borrowing the idea and success of functional Building Blocks (BB) into the neural data modeling. With a stacking of state-of-the-art techniques and domain knowledge, the proposed method achieves effectiveness through empirical evidence. However, these findings are a bit heuristic and empirical. There are few theoretical guarantees in this paper.

**Questions:**

1. How can you prove that the Building Blocks (BB) are capable of modeling the spatio temporal structures within the states and dynamics of neural data well than traditional methods like State Space Models (SSM) and Variational Autoencoders (VAE)?
2. How does the proposed method ensures to model and distinguish the within-state and between-state variabilities and relationships, which could be crucial for a comprehensive analysis of the data.
3. What are the bio-plausible insights of this paper and in the model design?

---

> ### Author Response · Authors · 2023-11-17
> **Response 1: Novelty, empirical results, and theoretical guarantees**
>
> We thank the reviewer for the valuable feedback on our paper and would like to address the concerns below:
>
> **Novelty:**
>
> The novelty of our work goes far beyond the application of a concept to neuroscience.
> It lies in the development of a versatile framework that identifies functional building blocks through re-weighted temporal graphs, fostering sparsity in multi-way data while considering within-state and between-state information.
>
> Our method tackles issues of real world data that other models typically overlook, such as varying trial durations, different trial counts per state, and handling of missing data or sampling rates, as well as enabling the discrimination between background and state-specific building blocks.
>
> The potential applications of this method extend well beyond neuroscience, with potential applications in genomics, consumer behavior, financial markets, environmental sciences, urban planning, and the study of social dynamics
> (For additional details, please refer to the examples provided in our second response to reviewer T8G6).
> We believe our approach is a standalone method with universal applicability, including contributions to neuroscience, as well as to other fields.
>
> **Empirical results and theoretical guarantees:**
>
> Our paper presents a comprehensive empirical exploration of SiBBlInGS, highlighting its adaptability and broad applicability through various synthetic and real-world examples. The approach is grounded in the theoretical principles of dictionary learning, drawing from seminal works in the field (e.g., [1], [2], [3], [4]). However, we acknowledge that a comprehensive theoretical discourse on dictionary learning is beyond the scope of our 9-page, application-focused paper.
>
> In particular, in [1], the authors provide a geometric characterization of the optimization landscape in dictionary learning, eliminating concerns over spurious local minimizers. [2] shows theoretical guarantees that detail how searching within an over-realized model space can foster the recovery of dictionaries, conceptually linking this process with generalization bounds and introducing an efficient strategy for extracting accurate model components. [3] establishes the theoretical framework necessary for confidently recovering dictionaries under sparsity constraints, and [4] marks a deep dive into sample complexity, stating how much data is needed to reliably engage in dictionary learning and parallel matrix factorization efforts.
>
> We will include these references with a brief explanation in the introduction of the updated paper to emphasize the above.
>
>
> [1] Sun, J., Qu, Q., & Wright, J. (2016). Complete dictionary recovery over the sphere I: Overview and the geometric picture. IEEE Transactions on Information Theory, 63(2), 853-884.
>
> [2] Sulam, J., You, C., & Zhu, Z. (2022). Recovery and generalization in over-realized dictionary learning. The Journal of Machine Learning Research, 23(1), 6060-6082.
>
> [3] Gribonval, R., & Schnass, K. (2010). Dictionary Identification—Sparse Matrix-Factorization via $\ell_1 $-Minimization. IEEE Transactions on Information Theory, 56(7), 3523-3539.
>
> [4] Gribonval, R., Jenatton, R., Bach, F., Kleinsteuber, M., & Seibert, M. (2015). Sample complexity of dictionary learning and other matrix factorizations. IEEE Transactions on Information Theory, 61(6), 3469-3486.

---

> ### Author Response · Authors · 2023-11-17
> **Response 2: Comparison to SSMs and VAEs and Distinguishing Within and Between State Variability**
>
> We thank the reviewer for raising concerns about potential similarity to SSMs and VAEs.
>
>
> **Comparison to SSMs and VAEs:**
>
> *SSMs:*
>
> In State-Space Models (SSMs), latent states evolve according to a transition matrix that dictates how the state evolves, accompanied by an emission matrix that determines how the latent state translates to the observations. The resemblance of SSMs to SiBBlInGS may not be immediately clear—whether it pertains to the structure of the emission matrix or the potential dynamics operators in latent space.
>
> If the concern was about potential similarities between SiBBlInGS and the emission matrix in SSMs, our approach incorporates re-weighted sparsity as a guiding principle, unlike SSMs' emission. Specifically, we utilize temporal-similarity-driven re-weighted $\ell_1$ regularization to find cross-state BBs, a feature distinct to SiBBlInGS. Furthermore, SSMs do not focus on studying cross-trial variability or on pinpointing the formation of emergent BBs under specific conditions.
>
> If the reference is to the potential analogy between the BBs and the transition operators of SSMs or other dynamical systems models (e.g., as seen in the identification of dynamical operators such as those in rSLDS [1] or dLDS [2]), SiBBlINGS find BBs that are not a function of time, and the temporal operators are not evolving by dynamical operators.  SiBBlInGS’ BBs act as sparse interpretable functional clusters of channels found based on co-activity with cross-state constraints, while dynamical operators in [1] or [2] are neither sparse nor integrating temporal-similarity regularization.
>
> *VAEs:*
>
> In the context of Variational Autoencoders, if the comparison is being drawn between SiBBlInGS' BBs and VAE's latent layer, it's essential to note that VAEs' latent layers are not inherently sparse and do not represent clusters from the original input (channels) space. They are found through sequential non-linear projections, resulting in a loss of interpretability and representing something different from SiBBlInGS' BBs. Even in the context of sparse VAEs (e.g., [3]), it's important to note that there is no co-learning of the components alongside their temporal traces. Additionally, with respect to dynamical VAEs (e.g., [4]),  these models not align with SiBBlInGS's aim of understanding multi-state variability.
>
> [1] Glaser, J., Whiteway, M., Cunningham, J. P., Paninski, L., & Linderman, S. (2020). Recurrent switching dynamical systems models for multiple interacting neural populations. Advances in neural information processing systems, 33, 14867-14878.
>
> [2] Mudrik, N., Chen, Y., Yezerets, E., Rozell, C. J., & Charles, A. S. (2022). Decomposed Linear Dynamical Systems (dLDS) for learning the latent components of neural dynamics. arXiv preprint arXiv:2206.02972
>
> [3] Ashman, M., So, J., Tebbutt, W., Fortuin, V., Pearce, M., & Turner, R. E. (2020). Sparse Gaussian process variational autoencoders. arXiv preprint arXiv:2010.10177.
>
> [4].Girin, L., Leglaive, S., Bie, X., Diard, J., Hueber, T., & Alameda-Pineda, X. (2020). Dynamical variational autoencoders: A comprehensive review. arXiv preprint arXiv:2008.12595.
>
>
> **Distinguishing Within and Between State Variability:**
>
> SiBBlInGS  adeptly integrates analysis both within and across states, maintaining consistency of the structure of within state BBs and accommodating nuanced controlled variations of between-state BBs. The degree of resemblance between corresponding cross-state BBs is methodically adjusted to reflect the corresponding levels of similarity between the states (as seen in the last term of Equation 1).

---

> > ### Author Response · Authors · 2023-11-17
> > **Response 3: Bio-plausible insights and  model design**
> >
> > We thank the reviewer for the comment on bio-plausible insights and model design.
> >
> > **Bio-plausible model design:**
> >
> > SiBBlInGS is informed by biologically plausible principles and is designed to uncover functional groups within real-world data, including biological data, based on several key features that align with biology:
> > - *Temporal Similarity-Based Functional Groups:*
> >
> > SiBBlInGS identifies functional groups rooted in temporal similarity, reflecting how some real-world entities like neurons
> >  functionally co-activate (e.g., as shown in [1,2,3]).
> >
> > - *Beyond Orthogonality Assumption:*
> >
> > The model challenges traditional orthogonal assumptions of components made, for instance, by PCA, HOSVD, and SRM [4], acknowledging that biological building blocks, like functional neural ensembles, may exhibit non-orthogonal interactions.
> >
> > - *Sparsity in Representations:*
> >
> > SiBBlInGS integrates sparsity, recognizing that in biological systems like the brain, functionally related groups are typically sparse subsets of the whole simultaneously recorded units.
> >
> > - *Handling States & Trials:*
> >
> > The model naturally handles data with multiple trials and several states, which is a common structure in real-world biological datasets. This is crucial for delineating variability across different conditions in scientific research, such as varying disease stages or comparisons between disease and control groups.
> >
> > - *Variable Trial Durations:*
> >
> > The model accommodates trials with variable durations, aligning with the variability seen in real time series data where the duration of events, like animal behavior tasks, is not necessarily fixed.
> >
> > - *Missing Data and  Sampling Rates:*
> >
> > SiBBlInGS is built to work with missing data points and varying sampling rates, addressing issues common in datasets ranging from clinical visits to temporary lapses in sensor measurements.
> >
> >
> > **Bio-plausible Insights:**
> >
> > In our paper, the emphasis was on showcasing the effectiveness of SiBBlInGS across various data types and highlighting its potential for advancing scientific discoveries, rather than being mainly focused on the biological insights.
> >
> > We do demonstrate though the model's potential in finding interpretable meaningful neural ensembles and in detecting subtle cross-state variations in these ensembles across states, along with informative traces. Specifically:
> > - *Functional Neuronal Groups in Motor Tasks:*
> >
> > For recordings from area 2 of the cortex, SiBBlInGS successfully discerned several neuronal groups engaged during a reaching task. The model found subtle adaptations of the cross-state neural BBs’ structures (Figure 4B), and showed that their temporal signatures can encode  the monkey’s decision-making process (Figure 4C).
> >
> > - *Seizure Detection in EEG Data:*
> >
> > The model proved its efficacy in an EEG data from an epileptic patient, where, unlike other methods, it identified the seizure onset location seconds prior to the event. This indicates SiBBlInGS's promise in epilepsy research.
> >
> >
> >
> > These results highlight the model's versatility and scientific utility, suggesting its potential for valuable insights in biological and  scientific research. We anticipate employing it in upcoming scientific projects for biological discoveries.
> >
> > [1] Gerstein, G. L., Perkel, D. H., & Subramanian, K. N. (1978). Identification of functionally related neural assemblies. Brain research, 140(1), 43-62.
> >
> > [2] Malik, A. N., Vierbuchen, T., Hemberg, M., Rubin, A. A., Ling, E., Couch, C. H., ... & Greenberg, M. E. (2014). Genome-wide identification and characterization of functional neuronal activity–dependent enhancers. Nature neuroscience, 17(10), 1330-1339.
> >
> > [3] Yuste, R., Nelson, D. A., Rubin, W. W., & Katz, L. C. (1995). Neuronal domains in developing neocortex: mechanisms of coactivation. Neuron, 14(1), 7-17
> >
> > [4] Chen, P. H. C., Chen, J., Yeshurun, Y., Hasson, U., Haxby, J., & Ramadge, P. J. (2015). A reduced-dimension fMRI shared response model. Advances in neural information processing systems, 28.

---

> ### Comment · Reviewer_WRaP · 2023-11-22
>
> I thank the author for the detailed response with insights.

---

> > ### Author Response · Authors · 2023-11-22
> >
> > Thank you for appreciating our response. We would like to inquire if there are any additional concerns on your end and if we adequately addressed the novelty issue, which appeared to be a primary concern from your perspective

---

### Official Review · Reviewer_xmKq · 2023-11-01

**Soundness:** 3 good
**Presentation:** 2 fair
**Contribution:** 3 good
**Rating:** 5
**Confidence:** 4

**Summary:**

The authors address the task on analysing high-dimensional time-series data.  Explicitly accounting for states and trials, they perfrom a per-state-and-trial matrix factorization. As part of this, they infer factor matrices which they term building blocks (BBs); similarity of BBs is controlled via a state-similarity graph.

**Strengths:**

In the problem setup, the authors define a setting where allowing for observations stemming from different states, and sessions, as well as allowing for different durations between states/trials.  This setup reflects real-world applications well and as such has received little attention in the literature.

**Weaknesses:**

- The authors compare their model only to very simple baselines, such as PCA and vanilla PARAFAC. In particular in the context of PARAFAC a lot of recent literature exists that generalized PARAFAC to explicitly account for temporal dependencies. Such approaches should be discussed explicitly and systematically benchmarked. In particular the authors should consider [1,2,3], where temporal information and different states over time are modelled via GP priors or parametric regularizers.
- The experiments are very limited. While there are some analyses on synthetic data, they should be extended to include more baselines and also demonstrate how the proposed Method works in different settings for high-dimensional time series data e.g. such as anslysed in [1]
- Results from baselines should also be discussed for the real-world applications
- I find the presentation of the paper could be improved: it requires a lot of jumping back and forth between appendix and main text for important results and to gain a good understanding.  I also found the results of the real world applications hard to understand.



[1] Wang, Z., & Zhe, S. (2022, June). Nonparametric Factor Trajectory Learning for Dynamic Tensor Decomposition. In International Conference on Machine Learning (pp. 23459-23469). PMLR.

[2] Tillinghast, C., Fang, S., Zhang, K., & Zhe, S. (2020, November). Probabilistic neural-kernel tensor decomposition. In 2020 IEEE International Conference on Data Mining (ICDM) (pp. 531-540). IEEE.

[3] Ahn, D., Jang, J. G., & Kang, U. (2021, October). Time-aware tensor decomposition for sparse tensors. In 2021 IEEE 8th International Conference on Data Science and Advanced Analytics (DSAA) (pp. 1-2). IEEE.

**Questions:**

See above

---

> ### Author Response · Authors · 2023-11-20
> **Response 1: Additional Baselines**
>
> We appreciate the reviewer's thoughtful feedback on our manuscript and the referral to additional sources and references.
>
> **Additional Baselines:**
>
> We appreciate the feedback from the reviewer. In response, we have provided a brief description of additional models in the ''BACKGROUND AND RELATED WORK" (see in blue font) to underscore the distinctions from SiBBlInGS.
>
> Furthermore, we added two more experiments, comparing our method also to: 1) NONFAT [1], and 2) NNDTN (discrete-time NN decomposition with nonlinear dynamics). The modifications include updates to Figure 2 in the paper to incorporate these comparisons, along with the addition of Figures 14 and 15, illustrating the identified components for all methods and the correlation with the ground truth per BB and per state. Please consult the updated paper for details on how we extracted these components, as described in the supplementary material.
>
> Broadly, SiBBlInGS serves a distinct purpose compared to methods mentioned in [1, 2, 3] or other Gaussian Process (GP)-based approaches. While these methods leverage temporal dependencies for robust decompositions with predictive power for future time points, SiBBlInGS aims to identify interpretable functional groups based on co-activity with subtle controlled variability along different axes—the trial and state axes, which are orthogonal to the time axis.
> In particular, SiBBlInGS focuses on discovering interpretable components with their traces rather than predicting future time points (as presented in [1] for instance) or solely reconstructing test data.
>
> Furthermore, the identified components in some of these other methods are not naturally sparse, and they do not aim to group sparse subsets of channels based on per-state temporal similarity. Hence, their interpretation does not align well with the discovery of functional groups that, in various fields, are assumed to represent co-active components, such as neural ensembles. Moreover, some of these methods cannot naturally work with data of varied duration or rates due to the inherent tensor structure.
>
> Although integrating GP ideas and dynamical processes into the analysis, as seen in [1, 2], is promising and could be very interesting for future work to better model within-trial non-stationary temporal conditions, it is not the primary goal of the current framework.
>
> [1] Wang, Z., & Zhe, S. (2022, June). Nonparametric Factor Trajectory Learning for Dynamic Tensor Decomposition. In International Conference on Machine Learning (pp. 23459-23469). PMLR.
>
> [2] Tillinghast, C., Fang, S., Zhang, K., & Zhe, S. (2020, November). Probabilistic neural-kernel tensor decomposition. In 2020 IEEE International Conference on Data Mining (ICDM) (pp. 531-540). IEEE.
>
> [3] Ahn, D., Jang, J. G., & Kang, U. (2021, October). Time-aware tensor decomposition for sparse tensors. In 2021 IEEE 8th International Conference on Data Science and Advanced Analytics (DSAA) (pp. 1-2). IEEE.

---

> > ### Author Response · Authors · 2023-11-20
> > **Response 2: Experiments and Presentation**
> >
> > We thank the reviewer for the comments regarding the experiments and presentation.
> >
> > **Experiments:**
> >
> > With respect to the concern about the experiment's scope, we've strived to maximize the presentation of our model's efficiency within the 9-page constraint for ICLR papers. The paper includes four experiments, each exploring different settings, characteristics, and conclusions. In response to the feedback and within the short rebuttal period, and as you suggested, we've added two more experiments, specifically comparing SiBBlInGS to NONFAT [1] and to NNDTN (discrete-time NN decomposition with nonlinear dynamics, a model mentioned in [1] as well). Yet, it is crucial to underscore the differences in both goal and assessment methodology between SiBBlInGS and the approach and the baselines presented in [1]. The objective of [1] and most of its baselines revolve around considering temporal non-stationarities through Gaussian process priors utilizing the frequency domain. In contrast, SiBBlInGS is designed to discover interpretable sparse BBs that represent functional groups based on similar activity, and seeks to understand how these groups vary across and within conditions, with changes occurring over trials—a different axis of focus compared to [1]. Moreover, as demonstrated in the cortex experiment, due to regularizations on $\Phi$—giving the model a high number of BBs as an hyperparameter (i.e. too large $p$) will result in non-active or noise-capturing BBs (Figure 4).
> >
> > Regarding the discussion of baseline results in real-world applications, we recognize the importance of comparisons. However, as the BBs of these real-world examples are unknown, and it is precisely what SiBBlInGS aims to reveal, quantitative comparisons  are intractable (given the unsupervised nature). Our approach has been to showcase SiBBlInGS' capability in recovering ground truth BBs in synthetic data, and then demonstrate its effectiveness  for the real-world examples (where ground truth BBs are unavailable) in providing interpretable, meaningful, sparse BBs with cross-state informative controlled variations.
> >
> > Additionally, for the EEG experiment, we present results from other baselines in Figure 13, where it is qualitatively evident that these models fall short in capturing the structural emergence of a BB around the area the seizure originates from. Unfortunately, for that experiment, a comparison with NONFAT and NNTDN is unfeasible, as these methods are based on Torch and are computationally expensive, rendering them impractical to run on our PCs during the rebuttal for this high-dimensional data—this, in itself, underscores another advantage of SiBBlInGS.
> >
> >
> > **Presentation:**
> >
> > Thank you for your feedback on the results presentation. We acknowledge the constraints of conference papers, which limit us to 9 pages. In response, we've prioritized clarity in the model description and highlighted various experiments showcasing different features. Supplementary details, considered less urgent, have been included for completeness. In light of your comment, we've considered enhancing the presentation flow by first introducing the inference for $A$ and only then presenting the kernels. Please find this modification in the attached updated paper.

---

> > > ### Author Response · Authors · 2023-11-20
> > > **Response 3: Understanding real world applications**
> > >
> > > We thank the reviewer for the comment about understanding the real-world applications.
> > >
> > > The real-world examples each serves a distinct goal and highlight specific features of SiBBlINGS. We are available to address any specific questions that may arise.
> > >
> > > In the Google Trends experiment, the aim was to identify groups of terms with similar search patterns, capturing both cross-state structural and temporal variability. Clear and interpretable BBs were discovered, exhibiting temporal traces aligned with relevant BB changes (e.g., increased searches during winter for the 'winter' BB). Additional structural variability between states was observed; for instance, the Passover BB showed higher temporal changes in states with higher percentages of Jewish populations. Similarly, searches for decorations were more prominent in New York as part of the 'winter' BB.
> > >
> > > The cortex experiment demonstrated the detection of neuronal ensembles based on shared cross-state temporal activity, revealing subtle controlled structural variations. These variations balance interpretability, ensuring that ensembles across states correspond to each other without being overly different, yet supporting small adaptations in ensemble compositions that indicate subtle changes in the membership magnitudes of neurons across task conditions (see Fig 4B). Additionally, intra-state temporal correlations were found to be larger than inter-state correlations, indicating that the model captures within-state versus between-state variability while providing an interpretable description (see Fig 4E). Moreover, the traces of the BBs proved informative for predicting task states (see Fig. 4C).
> > >
> > > In the epilepsy data experiment, we segmented the rich dataset into periods well before a seizure (normal activity), pre-seizure (over the 8 seconds before the onset), and during the CPS ("complex partial seizure"). One of the BBs SiBBlInGS found (the pink one) exhibited a pronounced structural change in the seconds preceding the seizure in a location that aligns with the clinically identified area from where the seizure originated—a change not captured by other methods (see Figures 13 and 5).

---

> > > > ### Author Response · Authors · 2023-11-23
> > > > **Kind Reminder - Response to Our Rebuttal**
> > > >
> > > > Dear Reviewer,
> > > >
> > > > we appreciate your thoughtful review of our paper. As the discussion period deadline approaches, we kindly request your response to our rebuttal and welcome any additional questions you may have. We have carefully addressed your concerns and incorporated additional comparisons based on your suggestions.
> > > >
> > > > SiBBlInGS stands out from other tensor factorization approaches by discovering interpretable similarity-driven sparse components in multi-state time series data while considering both temporal and state similarities to distinguish within and between-state variability. It accommodates trials with varying durations or sampling rates, addresses real-world scenarios with BBs exhibiting subtle cross-state structural variability and per-trial temporal variability, and is capable of distinguishing between state-specific and state-invariant BBs.
> > > >
> > > > We have considered your comments and addressed them in our rebuttal, along with an updated version of the paper. We look forward to hearing from you and thank you for your time and consideration.

---

> ### Comment · Reviewer_xmKq · 2023-11-23
>
> I thank the authors for the responses to my questions.

---

### Official Review · Reviewer_T8G6 · 2023-11-06

**Soundness:** 3 good
**Presentation:** 3 good
**Contribution:** 3 good
**Rating:** 6
**Confidence:** 2

**Summary:**

This paper presents a framework called SiBBlInGS, which stands for Similarity-driven Building Block Inference using Graphs across States. The framework is designed to discover fundamental representational units within multi-dimensional data, which can adjust their temporal activity and component structure across trials to capture the diverse spectrum of cross-trial variability. The paper discusses the limitations of existing methods for understanding multi-dimensional data and how SiBBlInGS addresses these limitations. It also explains how SiBBlInGS employs a graph-based dictionary learning approach for building block discovery, and how it considers shared temporal activity, inter- and intra-state relationships, non-orthogonal components, and variations in session counts and duration across states. Finally, the paper compares SiBBlInGS to other approaches for discovering fundamental representational units within multi-dimensional data and discusses potential applications of this framework in scientific domains.

**Strengths:**

- The SiBBlInGS framework is a novel approach for discovering fundamental representational units within multi-dimensional data. It addresses the limitations of existing methods and considers shared temporal activity, inter- and intra-state relationships, non-orthogonal components, and variations in session counts and duration across states.
- SiBBlInGS is designed to be resilient to noise, random initializations, and missing samples. This makes it a robust framework for discovering building blocks in real-world data.
- The paper includes a thorough evaluation of the SiBBlInGS framework on both synthetic and real-world data. The results demonstrate the effectiveness of the framework in discovering building blocks and its potential for applications in scientific domains.

**Weaknesses:**

- The proposed framework is complicated. While the paper provides a high-level overview of the SiBBlInGS framework, it does not provide detailed implementation instructions or code. This may make it difficult for researchers to replicate the results or apply the framework to their own data.
- While the paper discusses potential applications of the SiBBlInGS framework in scientific domains, it does not provide concrete examples of real-world applications. This may limit the impact of the framework and its adoption by researchers in different fields.
- The authors did not provide more details on the limitations of the SiBBlInGS framework and potential areas for improvement? This would help readers understand the scope and applicability of the framework.

**Questions:**

- Can the authors discuss the potential limitations of the SiBBlInGS framework in terms of scalability and computational efficiency? For example, how might the framework perform on larger datasets or in real-time applications, and what steps can be taken to address these limitations?
- How does the proposed method compare to other approaches for discovering fundamental representational units within multi-dimensional data?
- How does the shared temporal activity, inter- and intra-state relationships, and non-orthogonal components contribute to the discovery of building blocks and the effectiveness of the framework?
- Can the authors explain more on how SiBBlInGS handles missing samples and varied sampling rates in multi-dimensional data? How does it ensure that the discovered building blocks are robust to these variations?
- What advantages does graph-based dictionary learning approach offer different from other dictionary learning approaches?

---

> ### Author Response · Authors · 2023-11-17
> **Response 1: Model Complexity, Real-World Examples, Limitations, and Scalability**
>
> We appreciate the strengths highlighted by the reviewer, as well as the questions and feedback provided. We will now address the concerns and questions raised below:
>
> **Implementation Instructions and Code:**
> As presented in Algorithm 1 (page 14) as well as given the code that was attached in the supplementary materials, we believe the algorithm can be used by future researchers as well. Once accepted and dis-anonymized, a GitHub repository with the code will be shared and a Python PyPi package for simple pip installation will be created for simple usage of the algorithm and to enable users to develop custom implementations and modifications.
>
>
> **Examples for Real World Applications:**
> In addition to the real-world examples mentioned in the ''CONCLUSIONS AND DISCUSSION" section (section 6), please find below some more details about potential applications of SiBBlInGS:
> - *Consumer Products:* SiBBlInGS could identify groups of products (channels) with similar purchasing patterns over time (time) across
> countries (states), and explore how these patterns vary by location to inform personalized customer recommendations.
> - *Economic Analysis:* SiBBlInGS could be applied to cluster stock volume data (channels) over time (time) and across countries (states), to uncover patterns reflecting cross-country market trends and differences in investor activity.
> - *Climate Science:* SiBBlInGS could be used to cluster carbon dioxide emission patterns of various countries (channels) across daily intervals (time) and yearly changes (states). This analysis aids in discerning global emission trends and could inform more targeted climate policy initiatives.
> - *Social Dynamics Analysis:* SiBBlInGS could be utilized to analyze and understand the formation and evolution of online user groups (channels’ BBs) based on shared temporal patterns of user activity (time) across different states (e.g. before/ during / after the COVID-19 pandemic). This approach could enable the observation of shifts in social interactions and group dynamics in response to global events.
> - *Genetics Research:* SiBBlInGS could analyze multi-tissue gene expression data (e.g., as the data described in [1]) to identify functional gene groups (channels) and assess how their expression across tissues (~time) differs between individuals (states).
>
> [1] Hore, V., Vinuela, et al. (2016). Tensor decomposition for multiple-tissue gene expression experiments. Nature genetics, 48(9), 1094-1100.
>
> **Potential areas for improvement:**
> As outlined in the final paragraph of the paper, certain limitations, beyond the current scope of the paper, will be addressed in future work. In particular, currently SiBBlInGS presumes normal noise distribution (e.g., as seen in the $\ell_2$ norms of the data fidelity terms in equations 1,2) and Euclidean distance between state labels in the supervised approach (definition of $P$), which may not align well with some data (e.g., neural data may be modeled with a Poisson distribution in low spike count rates). Additionally, while the identified BBs capture channel membership and magnitude, they currently do not account for potential directed connectivity within the BBs, presenting an opportunity for future extensions.
> For future work, we also plan to showcase SiBBlInGS's versatility by integrating diverse data modalities with the same structure (e.g., fMRI vs ultrasound of the same brain area). Additionally, we plan to extend its usability to handle a continuous, infinite number of non-discrete labels, as well as incorporating extra axes for state changes, allowing a trial to belong to several states within a pool of states.
>
>
>
> **Potential limitations and improvements of SiBBlInGS in terms of scalability and computational efficiency:**
> As shown in Section E on model complexity, the current implementation's speed scales with the dataset's dimensionality, encompassing the number of channels, states, time points, and the rank of each trial. To enhance computational speed, the implementation already incorporates random subsampling of time periods and trials within each state during iterations (as mentioned in the paper after equation 1). This allows the model to process only a portion of the data at each update step, eventually exposing the model to all trials and time points. Additionally, the current implementation offers users to choose between different lasso solvers that vary in speed. Future improvements for efficiency may involve implementing the model in languages other than Python, such as C++ or Julia, and applying parallel computing for faster and more efficient performance.
> With respect to real-time applications, SiBBlInGS is not intended for real-time inference. However, if BBs are found previously based on some training data for a state, they can be easily used in real-time applications to find their respective traces when new observations arrive (i.e., by solving equation 9).

---

> ### Author Response · Authors · 2023-11-17
> **Response 2: Comparing SiBBlInGS to Other Approaches, Significance of Temporal Activity, Inter/Intra-State Relationships, and Non-Orthogonality.**
>
> We appreciate the reviewer's comments on SiBBlInGS compared to other approaches and its effectiveness.
>
> **Comparing SiBBlInGS with Other Approaches:**
>
> As demonstrated in the synthetic data results (Figure 2), EEG experiments (Figure 13), and trends experiments (Figure 8), along with additional comparisons detailed in Table 2, we assessed our method against alternative tensor and matrix factorization techniques, through both theoretical analysis and practical experiments.
>
> SiBBlInGS stands apart from classic tensor factorization methods, as it finds similarity-driven sparse BBs with subtle cross-state variations, accommodates variable trial durations and inherently incorporates state information and similarities into the analysis, effectively differentiating intra- and inter-state variability— aspects not addressed by other approaches. Additionally, certain methods presume orthogonality among components, a condition that may not hold true for some real-world cases (e.g.,  neural ensembles). These alternative approaches often tend to find components based on maximum variance or energy rather than co-activity, and they are not inherently tailored to differentiate state-specific from state-invariant building blocks (as $\nu$ does in SiBBlInGS).
>
> **Significance of Temporal Activity, Inter/Intra-State Relationships, and Non-Orthogonality:**
>
> *Shared Temporal Activity:*
>
> We appreciate the reviewer’s question about the contributions of different features of SiBBlInGS to the framework.
> As detailed following Equation 1 in the paper (starting with 'The weighted evolving regularization terms for each state...'), the channel similarity graph (${H}$) is built based on temporal activity similarities between channels per trial. This graph is integrated into the denominator of the regularization weight ($\lambda$) in the re-weighted lasso equation (2nd part of Equation 1) and thus fosters the aggregation of channels with co-active patterns into the same BBs.
>
> In particular, the term
> $H_{n:}A_{:j}$
> that appears in the denominator for updating $\lambda_{n,j}^d$ (2nd formula in Equation 1) reflects the correlation between the activity of the $j$-th BB (as defined by $A_{:j}$) with the similarity levels of all channels to the $n$-th channel (as defined by $H_{n:}$).
> Since that term appears in the denominator of $\lambda_{n,j}^d$—a higher correlation between $A_{:j}$ and $H_{n:}$ (which implies that the temporal neighbors of the $n$-th channel might be in the $j^{th}$ BB), results in a smaller sparsity weight ($\lambda_{n,j}^d$ decreases)—thus promoting that $n^{th}$  channel's inclusion in the $j^{th}$ BB and effectively ‘’pushing” the  channel to the BB that includes its temporal neighbors.
>
> Conversely, if the temporal neighbors of channel $n$ mainly do not belong to that $j$-th BB, the term  $H_{n:}A_{:j}$ will be smaller, hence $\lambda_{n,j}^d$ is larger, resulting in an increased sparsity regularization that effectively pushing that $n$-th channel out of the $j^{th}$ BB.
>
> Another effect of the shared temporal activity is expressed in the data-driven approach, as the state graph is a function of the temporal similarities between states, and the regularization on the distance between cross-state BBs is proportional to the temporal similarities between these states (stored by $P$, and as seen in Equation 1), thereby encapsulating state-temporal similarities within the analysis.
>
> Specifically, last term in Equation 1 ($P_{dd'} \|\|a_{i:} - a_{j:}\|\|_2^2$) modulates the $\ell_2$ norm on the distance between corresponding BBs across different states in accordance with the states' similarities, ensuring that more similar states are associated with more similar BB patterns.
>
>
>
> *Non-orthogonal components:*
>
> Our method's ability to identify components without enforcing orthogonality constraints, as opposed to techniques like HOSVD or PCA, aligns better with real-world data where such constraints may not naturally exist. For example, in the context of neural ensembles, it is plausible to encounter functional groups that are non-orthogonal, both structurally and temporally.
>
>
>
> *Intra vs inter state relationships:*
>
> As written just before Equation 1 (the paragraph starting with “Assuming consistent underlying groups between same-state trials,....”), the BBs are kept constant between trials of the same state while supporting subtle controlled structural variations between states, thus addressing both resolutions. Unlike the regularized BBs, SiBBlInGS allows for flexible variations in temporal traces within and between states, capturing the complete spectrum of activity variability while maintaining a shared basis of BBs for interpretability. Additionally, as shown in Figure 4E, the within-state temporal trace correlations are larger than between-states temporal correlations of corresponding ensembles, indicating reduced within state compared to between states variability.

---

> > ### Author Response · Authors · 2023-11-17
> > **Response 3: Handling Missing Samples, Varying Sampling Rates, and Graph-Based Dictionary Learning**
> >
> > **Handling missing samples and varied sampling rates:**
> >
> > We appreciate the reviewer's question regarding the treatment of missing samples and varied sampling rates. They are addressed in slightly different dimensions:
> >
> > *Varying Sampling Rates:*
> >
> > SiBBlInGS adeptly integrates data from trials or states, even when acquired from machines with varying sampling rates—a prevalent challenge in real-world data analysis. The model's channel similarity graph is constructed based on per-trial similarities, ensuring that cross-trial non-simultaneous observations or cross-trial observations with varied rates pose no hindrance.
> >
> > In its supervised option, SiBBlInGS uses the state labels to build the state similarity graph ($P$), thus independent from the sampling rate of different trials.
> >
> > In the inference phase, regularization between trials is exclusively applied to the BBs, which are not time-dependent and, consequently, unaffected by sampling rates. In contrast, the optimization for temporal traces (Equation 2) does not integrate cross-trial similarities, thereby accommodating variations in sampling rates.
> >
> >
> > *Missing Data:*
> >
> > (1) When encountering missing samples, SiBBlInGS can simply exclude the missing points during the graph calculation process, and (2) it omits their impact in the data fidelity term during the optimization phase.
> >
> > In particular, let $\Pi$ be a projection operator that receives two matrices of the same dimension as inputs—a ''value" matrix (X) and a mask matrix (Y), such that if $Y_{ij} = 0$ , then $\Pi {(X,Y)}_{i,j} = 0$,
> >
> > and $\forall i,j$ s.t. $Y_{ij} \neq 0$, then
> >
> > $\Pi(X,Y)_{i,j}$
> >
> >  = $ {X}_{ij}$.
> >
> > Consider the above projection applied to the data fidelity term during the optimization (e.g., Equation 2), before applying the Frobenius norm: $|| \Pi (Y_m^d - A^d \phi^T , Y_m^d) ||_F^2$—specifically, this description ensures that the error in the data fidelity term does not take into account missing entries, using the mask from the observations $Y_m^d$.
> >
> >
> > We also note that we empirically showed on the synthetic data that the model is robust to missing samples, noise, and initializations (Figures 2 and 6).
> >
> >
> > **Advantages of graph-based dictionary learning:**
> >
> > As detailed in the paragraph following Equation 1 (starting with 'The weighted evolving regularization terms for each state...'), the graph-enhanced re-weighted $\ell_1$ regularization differs from standard $\ell_1$ in that it incorporates temporal-similarity-driven weights that influence the sparsity of the BBs. This approach effectively reshapes the sparsity pattern of the BBs, ensuring not only sparsity but also grouping together channels with similar activity based on the channel graph $H$ that is integrated into the regularization. Additionally, the state graph $P$ also plays an important role by imposing an extra constraint, this time on cross-state BBs' similarity, making it proportionate to the similarity of the respective states. This results in an inference that is better aligned with the spatio-temporal information within the data compared to classical dictionary learning.

---

> > > ### Comment · Reviewer_T8G6 · 2023-11-22
> > >
> > > Thank the authors for the responses to my questions.

---

> > > > ### Author Response · Authors · 2023-11-22
> > > >
> > > > Thank you for acknowledging our response. We appreciate your feedback and would like to inquire if there are any specific aspects that remain a concern

---

### Author Response · Authors · 2023-11-20
**General Response Highlights: Novelty, Baseline Comparisons, Graph-Based Dictionary Learning, Model Instructions, Bio-Plausible Design, Applicability, and Overall Significance**

We thank the reviewers for their valuable feedback, and we have addressed each concern in the individual responses. We highlight below some of the important points:

**Novelty:** Regarding the novelty and significance of our framework, it's critical  to note that SiBBlInGS diverges from other tensor factorization methods. Other methods not only often assume uniform sampling rates and trial durations, but cannot support controlled state-driven cross-trial structural adaptations of underlying sparse components—hindering the study of cross-trial variability. In contrast,  SiBBlInGS not only captures changing trial duration and rates, but explicitly aims to discover sparse structures with controlled cross-trial variations while integrating state and time information.

**Additional comparisons:** In response to Reviewer xmKq's suggestions, we incorporated two additional baselines for the synthetic data experiments:, 1) NONFAT [1], and 2) NNDTN (see Figures 2,14,15). We would like to also emphasize the comparisons made for the real-world EEG example to baselines, as depicted in Figure 13.

**Advantage over regular dictionary learning:** Our graph-based dictionary learning approach offers a distinct and advantageous perspective compared to classical dictionary learning, particularly for identifying interpretable co-actively driven sparse BBs. As described in the paragraph following Equation (1), the graph-filtered re-weighted $\ell_1$ algorithm deviates from standard regularization by incorporating temporal-similarity-driven weights that influence the BB sparsity. This method ensures sparsity and furthermore groups together channels with similar temporal activity  via the similarity graph adjacency matrix ($H$ that is integrated into the regularization). Additionally, the state graph adjacency matrix ($P$) plays a crucial role in imposing extra constraints on cross-state BB similarity based on the similarity of their respective states. These graph regularizations result in solutions that better align with the data’s temporal cross-state information.

**Model instructions:** Reviewer T8G6 expressed concerns regarding model instructions. We emphasize the inclusion of the code in the supplementary material and refer to Algorithm 1 on page 14 for implementation guidance.

**SiBBlInGS follows bio-plausible model design:** Our modeling choices align with real-world data, particularly in biology. SiBBlInGS identifies functional groups based on temporal similarity, mirroring entities like neural ensembles. In contrast to methods such as PCA, HOSVD, and SRM [2], SiBBlInGS removes traditional orthogonal assumptions, acknowledging non-orthogonal interactions prevalent in biological building blocks (e.g., neural ensembles). Moreover, SiBBlInGS integrates sparsity, aligning with our understanding of biological functional groups (e.g., gene groups), wherein functionally related groups are typically sparse subsets. In addition, SiBBlInGS naturally handles trials and states, accommodating real-world variability often observed in biological datasets. Additionally, SiBBlInGS adapts to variable trial durations, reflecting the dynamics seen in real-time series data. SiBBlInGS is robust to missing data and varying sampling rates, addressing common issues in diverse real world datasets (e.g., missing clinical visits; temporary lapses in sensor measurements).

**SiBBlInGS is applicable to various fields:** While our paper focuses on demonstrating SiBBlInGS applicability to real-world search data, cortex activity, and EEG data,  we emphasize that SiBBlInGS is a versatile method applicable across diverse scientific disciplines. For instance, in genomics, SiBBlInGS could analyze multi-tissue gene expression data, identifying, e.g., functional gene groups and assessing inter-individual differences in expression across tissues. In consumer products, SiBBlInGS could  pinpoint product groups with similar purchasing patterns, aiding in personalized recommendations. In economics, SiBBlInGS may prove valuable for clustering stock volume data, e.g., revealing cross-country market trends and variations in investor activity. In climate science, SiBBlInGS could identify carbon dioxide emission patterns underlying global emission trends for targeted climate policy initiatives. In social dynamics, SiBBlInGS could unravel the formation and evolution of online user groups based on shared user engagement patterns, offering insights into societal shifts in response to global events.

Overall, we believe that SiBBlINGS is a standalone method for finding meaningful components in multi-way high-dimensional data across trials and states. It overcomes several challenges posed by existing models and aligns with the complexities of real-world data.


[1] Wang, Z., & Zhe, S. (2022). Nonparametric Factor Trajectory Learning for Dynamic Tensor Decomposition. ICML.

[2] Chen, P. H. C., et al.. (2015). A reduced-dimension fMRI shared response model. NIPS.

---

### Meta-Review · Area_Chair_8vUT · 2023-12-13

**Metareview:**

This paper is concerned with the problem of finding fundamental representational units from data collected under multiple distinct states. It presents a framework called SiBBlInGS, standing for Similarity-driven Building Block Inference using Graphs across States. It can adjust the blocks' temporal activity and component structure across trials to capture the diverse spectrum of cross-trial variability. The paper discusses the limitations of existing methods for understanding multi-dimensional data under the considered scenario and how SiBBlInGS addresses these limitations. It also explains how SiBBlInGS employs a graph-based dictionary learning approach for building block discovery, and how it considers shared temporal activity, inter- and intra-state relationships, non-orthogonal components, and variations in session counts and duration across states.

The considered setting is well motivated and expected to be able to address some real problems. However, two issues should be addressed for the paper to be published. First, as one reviewer asked, "how can you prove that the Building Blocks (BB) are capable of modeling the spatio temporal structures within the states and dynamics of neural data well than traditional methods like State Space Models (SSM) and Variational Autoencoders (VAE)?" To me, this is an important question to answer, in order for the identified building blocks to be clearly interpretable. Second, the current experiments are very limited.

**Justification For Why Not Higher Score:**

Two issues should be addressed for the paper to be published. First, as one reviewer asked, "how can you prove that the Building Blocks (BB) are capable of modeling the spatio temporal structures within the states and dynamics of neural data well than traditional methods like State Space Models (SSM) and Variational Autoencoders (VAE)?" To me, this is an important question to answer, in order for the identified building blocks to be clearly interpretable. Second, the current experiments are very limited.

**Justification For Why Not Lower Score:**

The considered setting is well motivated and expected to be able to address some real problems.

---

### Decision · Program_Chairs · 2024-01-16

Reject